# Anatomical basis and physiological role of cerebrospinal fluid transport through the murine cribriform plate

Jordan N Norwood[1]*, Qingguang Zhang[2], David Card[3], Amanda Craine[4], Timothy M Ryan[5], Patrick J Drew[2,4,6]*

[1]Cellular and Developmental Biology Graduate Program, Pennsylvania State University, University Park, United States; [2]Department of Engineering Science and Mechanics, Pennsylvania State University, University Park, United States; [3]Department of Physics, Pennsylvania State University, University Park, United States; [4]Department of Biomedical Engineering, Pennsylvania State University, University Park, United States; [5]Department of Anthropology, Pennsylvania State University, University Park, United States; [6]Department of Neurosurgery, Pennsylvania State University, University Park, United States

**Abstract** Cerebrospinal fluid (CSF) flows through the brain, transporting chemical signals and removing waste. CSF production in the brain is balanced by a constant outflow of CSF, the anatomical basis of which is poorly understood. Here, we characterized the anatomy and physiological function of the CSF outflow pathway along the olfactory sensory nerves through the cribriform plate, and into the nasal epithelia. Chemical ablation of olfactory sensory nerves greatly reduced outflow of CSF through the cribriform plate. The reduction in CSF outflow did not cause an increase in intracranial pressure (ICP), consistent with an alteration in the pattern of CSF drainage or production. Our results suggest that damage to olfactory sensory neurons (such as from air pollution) could contribute to altered CSF turnover and flow, providing a potential mechanism for neurological diseases.
DOI: https://doi.org/10.7554/eLife.44278.001

*For correspondence:
jnn120@psu.edu (JNN);
pjd17@psu.edu (PJD)

**Competing interests:** The authors declare that no competing interests exist.

## Introduction

The brain has a very high metabolic rate, but lacks a conventional lymphatic system for clearance of metabolites and waste products. It is thought that the movement of cerebrospinal fluid (CSF) plays an important role in removing these waste products (*Simon and Iliff, 2016*). CSF is constantly produced by the choroid plexus (*Damkier et al., 2013*), circulates through the subarachnoid space and into the brain via a periarterial route (*Iliff et al., 2012*), and then exits the brain via several routes (*Weed, 1923*; *McComb, 1983*; *Zakharov et al., 2004*; *Pollay, 2010*). Normal aging is accompanied by decreases in CSF production and increases in CSF outflow resistance (*May et al., 1990*; *de Leon et al., 2017*; *Czosnyka et al., 2004*). CSF turnover is disrupted in pathological conditions such as Alzheimer's disease (AD) and normal pressure hydrocephalus (NPH) (*Albeck et al., 1998*; *König et al., 2005*; *Silverberg et al., 2006*). The balance between CSF production and outflow plays a key role in setting and maintaining the intracranial pressure (ICP) (*Kosteljanetz, 1987*). Because CSF production is actively regulated to maintain normal ICP (*Marmarou et al., 1975*), damage to the outflow pathways could drive compensatory decreases in CSF production and turnover (*Czosnyka et al., 2004*) or altered patterns of CSF flow. Either of these could result in decreased waste clearance, as is seen in aging and some neurological diseases.

While there are lymphatic vessels in the meninges (*Aspelund et al., 2015*; *Louveau et al., 2015*), there is evidence in both humans and other mammals pointing to drainage of the CSF through the cribriform plate (CP) (*Zakharov et al., 2004*; *de Leon et al., 2017*; *Walter et al., 2006*; *Ma et al., 2019*; *Bradbury et al., 1981*; *Erlich et al., 1986*; *Yamada et al., 1991*; *Cserr and Knopf, 1992a*; *Weller et al., 1992*; *Kida et al., 1993*; *Boulton et al., 1996*; *Johnston et al., 2004*). The CP is a fenestrated bony plate of the ethmoid bone that separates the cranial and nasal cavities. Once through the plate, CSF is absorbed by lymphatic vessels in the nasal mucosa and drained into the cervical lymph nodes (*Bradbury and Cole, 1980*). There has been speculation that interstitial fluid (ISF) and CSF leave the brain via the extracellular space between olfactory sensory nerve (OSN) axon bundles (*Szentistvanyi et al., 1984*), as the intercellular space between axon bundles provide low-resistance directed pathways for fluid flow (*Syková and Nicholson, 2008*). Acute blockage of CSF outflow by surgically obstructing the CP results in an increase in resting ICP (*Mollanji et al., 2002*) and outflow resistance (*Silver et al., 2002*), providing evidence for a drainage role of the trans-CP pathway. Intriguingly, there may be a connection between the patency of the clearance pathway through the CP and neurodegenerative diseases (*de Leon et al., 2017*; *Ethell, 2014*). The neurons whose axons make up the nerve bundles that traverse the CP, OSNs, are exposed to environmental toxins (such as air pollutants [*Ajmani et al., 2016*]), which epidemiological studies have associated exposure to neurodegenerative diseases. Furthermore, anosmia and decreased acuity in the sense of smell, which will result from OSN damage, reliably precede many neurological disorders (*Ajmani et al., 2016*). Thus, damage to or decreases in the numbers of the OSN axons could increase the resistance to CSF outflow, triggering a downregulation in CSF production and turnover. However, a method to maintain the three-dimensional structure of the hard, calcified plate and traversing axons and soft tissues to visualize the microscopic anatomical basis of this movement has been lacking. To better understand the role of the CP CSF outflow pathway, we characterized the cellular anatomy of the structures that traverse the CP and explored the chronic effects of OSN ablation on CSF turnover. We also found that OSN ablation disrupted CSF drainage through the CP, but normal ICP was maintained.

## Results

### Macroscopic structure of the cribriform plate

We first examined the macroscopic anatomical structure of the mouse CP to elucidate any stereotyped anatomical structure of the calcified tissue. While the anatomy of the CP has been studied in humans (*Kalmey et al., 1998*) and other large mammals, both extinct and extant (*Pihlström et al., 2005*; *Bhatnagar and Kallen, 1974*; *Bird et al., 2014*; *Bird et al., 2018*), the structure of the mouse CP has not been as well characterized. To visualize the 3D anatomy of the calcified bone of the CP, skulls of adult mice (3 Swiss Webster and 3 C57BL/6J) were de-fleshed by dermestid beetles, and imaged with micro-CT. Reconstructed CT images from an example mouse are shown in *Figure 1A–C*, showing the typical morphology of the murine CP. Photographs of the CP after removing the caudal portion of the skull showed similar structure (*Figure 1D*). The side of the plate adjacent to the olfactory bulb was not planar, but rather convex to accommodate the rounded shape of the anterior ventral face of the olfactory bulbs (*Figure 1E*); consequently, thin sections (like those obtained using traditional histology) will only contain a fraction of the structure. We observed a calcified ridge along the midline of the CP (crista galli) with an irregular perforation pattern, with many small (<200 μm diameter) foramina (holes). While the positioning, number, and size of the smaller foramina were variable across mice, there were four major foramen appearing in the same locations in all mice imaged. Bilateral foramina were located on the anterior aspect of the plate, lateral to the crista galli, and midway along the dorsal ventral axis of the CP on each side (*Figure 1C,D,F*). Their location and structure were consistent across all animals examined with micro-CT. To quantify the total area of all the foramina (see Materials and methods), maximum intensity projections were made along the axis perpendicular to the face of the plate (*Figure 1F*). We found the total area of the foramina was $1.88 \pm 0.29$ mm$^2$ (mean $\pm$ std, n = 6, pooling both C57BL/6J and Swiss Webster mice) and the total area of the plate was $9.76 \pm 1.12$ mm$^2$ (mean $\pm$ std, n = 6, pooling both C57BL/6J and Swiss Webster mice), similar to previous reports (*Bird et al., 2018*). CT images with C57BL/6J and Swiss Webster mice showed similar CP morphology (*Figure 1—figure supplement 1A–D*), suggesting that for

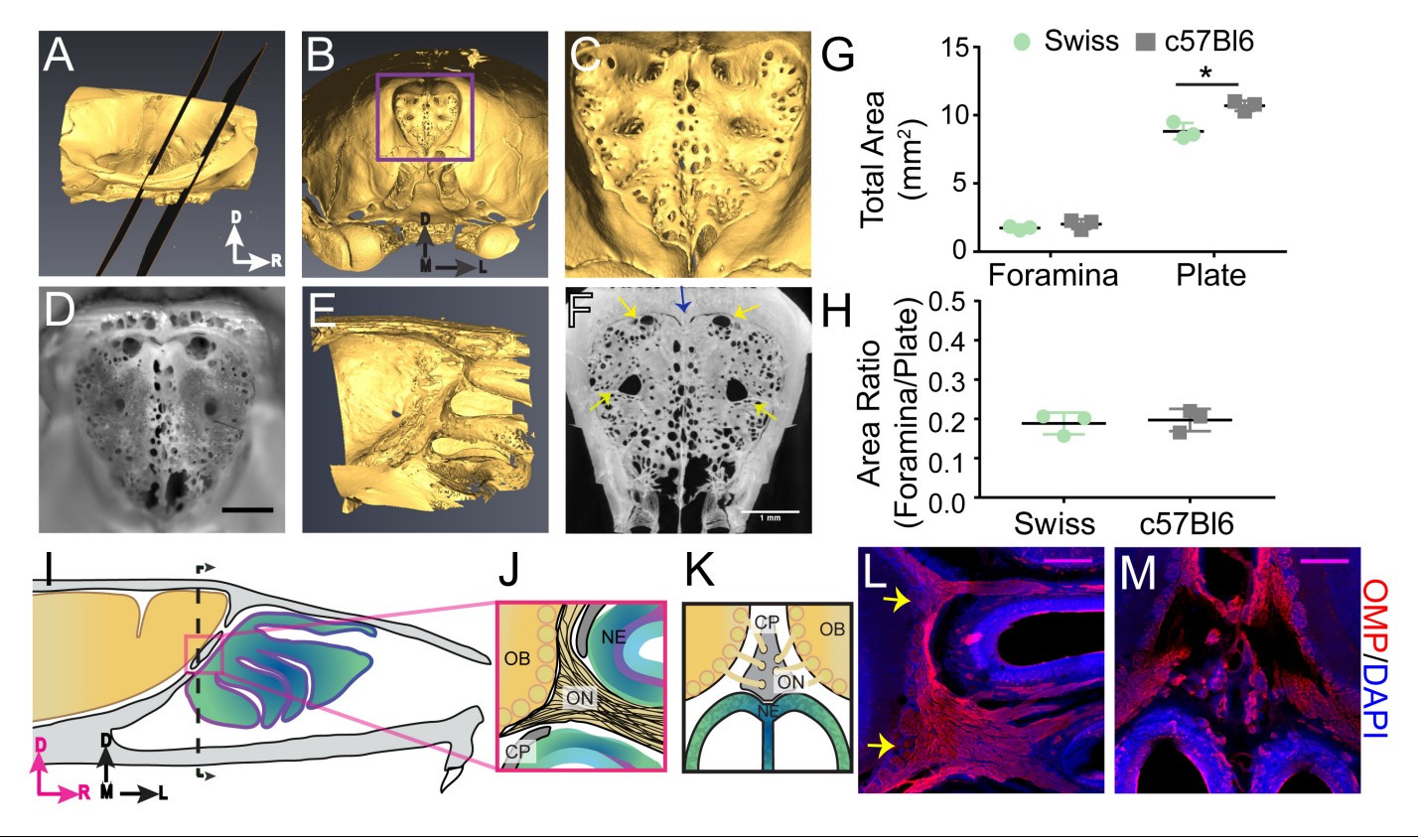

**Figure 1.** Structure of the calcified tissue of the cribriform plate and its relationship to olfactory sensory neuron axons. For schematics: olfactory nerve (ON), neuroepithelium (NE), cribriform plate (CP), olfactory bulb (OB) and glomeruli (yellow circles). D = dorsal, R = rostral, M = medial, and L = lateral. (A) Sagittal view of a microCT scan of a mouse skull. Black planes bracket the area of the CP. (B) Anterior-looking view of the CP from microCT image. (C) Area indicated by purple box in (B). (D) Photograph of the CP from the same point of view as the CT reconstruction in (C). Scale bar 1 mm. E) Sagittal view of the OB and CP junction illustrating the curved structure of the CP. (F) Max intensity projection of a 1.67 mm thick section of the CP depicting the major foramina (yellow arrows) and crista galli (blue arrow). Scale bar 1 mm. (G–H) Mean ± standard deviation plotted. Circles and squares represent means of individual animals. (G) Left: Comparison of total foramina area between Swiss Webster and C57BL/6J mice: (t(4) = 1.32, p=0.256, n = 3 for each group, ttest2). Comparison of the total area of the CP between Swiss Webster and C57BL/6J mice: (t(4) = 4.55, p=0.021, n = 3 for each group, ttest2). (*p≤0.05). (H) Comparison of the ratio of the foramina area and CP area between Swiss Webster and C57BL/6J mice: (t(4) = 0.3891, p=0.7170, n = 3 for each group, ttest2). (I) Schematic of the sagittal plane of the mouse skull and brain showing the relationship of the OB and nerve junction to the CP. (J) Sagittal view of the area within the pink box in (I), depicting OSNs crossing the CP and terminating in the OB and glomeruli. (K) Coronal view of the black dashed line in (I) illustrating the location of the CP relative to the OBs and NE. (L–M) Immunofluorescent staining, OMP (red) and DAPI (blue). Scale bars 250 µm. (L) Sagittal plane, area depicted in (J), showing the two main OSN axon bundles (yellow arrows) that pass through the major foramina of the CP. (M) Coronal plane, area indicated in (K), showing OSN axon bundles that traverse the minor foramina of the CP along the crista galli.

DOI: https://doi.org/10.7554/eLife.44278.002

The following figure supplement is available for figure 1:

**Figure supplement 1.** Cribriform plate morphology in Swiss Webster and C57BL/6J mice.
DOI: https://doi.org/10.7554/eLife.44278.003

these two strains, there are minimal strain differences. While a significant difference in total plate area (*Figure 1G*) was observed between C57BL/6J (10.693 ± 0.374 mm²) and Swiss Webster mice (8.821 ± 0.607 mm²), no significant difference was observed for total foramina area (*Figure 1G*) or in the ratio of foramina area to plate area (*Figure 1H*). Because the bone is impermeable to fluid movement, these holes will be the only outlet for CSF flow through the CP, and the nature of the soft tissue filling them (olfactory nerves, blood vessels, or lymphatic vessels) will play an important role in directing the flow of CSF.

## Olfactory nerves enter through the primary foramina

We then asked which foramina in the CP were traversed by OSN axons. However, the junction of the olfactory bulb and nerves is difficult to visualize histologically since the calcified CP, which the OSNs pass through, is immediately adjacent to the soft brain. To maintain the spatial alignment between the soft tissue and CP during sectioning, the bones of the skull must first be softened by decalcification. Traditional decalcification techniques require soaking the tissue in EDTA for several weeks (*Al Kawas et al., 1996*; *Naruse and Ueta, 2002*), which results in degradation of epitopes, making it incompatible with immunohistochemistry. We developed a rapid decalcification protocol using formic acid (see Materials and methods), allowing us to decalcify the skull within 2 days, thereby preserving epitopes for immunohistochemical labeling. Using formic acid decalcification, we were able to section the skull in a way that maintained the pathway of the OSNs (labeled with an anti-olfactory marker protein (OMP) antibody [*Buiakova et al., 1996*; *Ekberg et al., 2011*]) traversing the CP (*Figure 1L–M*). OMP-expressing OSN axon bundles were observed traversing the CP and synapsing onto the olfactory bulb (*Figure 1L,M*), with two major axon bundles (yellow arrows) on the left and right sides (*Figure 1L*). These OSN axon bundles filled the four major foramen observed in the CP (*Figure 1C,D,F*). In the coronal plane, smaller axon bundles traversed the CP through the smaller foramina bordering the crista galli (*Figure 1M*). These results show that there are two distinct types of foramen in the CP: the major foramen, through which the larger OSN bundles pass through, and a multitude of smaller holes whose position varied across animals and some of which carried smaller OSN bundles. Because extracellular fluid flow preferentially moves parallel to axon tracts (*Syková and Nicholson, 2008*), these nerves can provide directed conduits for CSF to move from the cranial cavity into the nasal cavity.

## Blood vessels traverse the cribriform plate

We then asked what other soft tissues were contained within the small foramen, as the nature of these tissues will influence the flow of fluid out of the cranial compartment. The presence of blood and lymphatic vessels in the foramina were assessed because both tissue types could be conduits for fluid flow (*Iliff et al., 2012*; *Aspelund et al., 2015*; *Louveau et al., 2015*).

We first looked at whether blood vessels traversed the CP by either filling the vessel lumen with FITC-albumin (*Tsai et al., 2009*), or labeling the endothelial cells by perfusing the vasculature with the fluorescent lipophilic dye DiI (*Li et al., 2008*). Note that we cannot distinguish between arteries and veins using these methods. FITC-albumin filled vessels were found to run parallel to the OSNs traversing the CP (*Figure 2D–E*). Blood vessels, labeled by DiI, were also clearly visible in the smaller foramina (*Figure 2M–O*). Both olfactory nerve bundles and blood vessels were present in other foramina (*Figure 2M*). These results are consistent with previous work showing blood vessels traverse the CP in the rodent (*Lochhead and Thorne, 2012*), much like the ethmoidal arteries and veins observed in humans (*Lochhead and Thorne, 2012*; *Knudsen et al., 1989*; *Yang et al., 2009*; *Souza et al., 2009*; *Tsutsumi et al., 2016*; *Tsutsumi et al., 2019*), linking the vascular territories of the nasal epithelia and brain.

## Aquaporins at the olfactory nerve-bulb interface

Aquaporins (AQP) greatly enhance the permeability of cells to water (*Takata et al., 2004*), and their presence would help facilitate the movement of fluid down the pressure gradient. The different aquaporin protein subtypes have similar structure, but differ in their localization throughout the body (*Takata et al., 2004*). We used immunofluorescence to examine the expression of aquaporin channels at the interface between the CP and olfactory bulbs. We validated each antibody using mouse kidney tissue, except the AQP4 antibody was validated in the cortex (*Figure 2—figure supplement 1L–O*) (*Verkman et al., 2017*; *Sakai et al., 2014*; *Procino et al., 2011*). We first looked at the expression of aquaporin-1 (AQP1), as it has been shown that expression of AQP1 in the CNS begins early (*Johansson et al., 2005*; *Weller et al., 2018*), and is required in adulthood to maintain CSF production by the choroid plexus (*Oshio et al., 2005*). Interestingly, previous work has also shown that AQP1 is expressed along the periphery of the olfactory bulb in neonatal mice (*Shields et al., 2010*), and there are high levels of expression of AQP1, 3, and 5, within the nasal cavity (*Ablimit et al., 2006*). The junction of the olfactory bulb and nerve is a prime exit route for CSF into the nasal cavity. Expression of aquaporins in this area would facilitate the flow of fluid

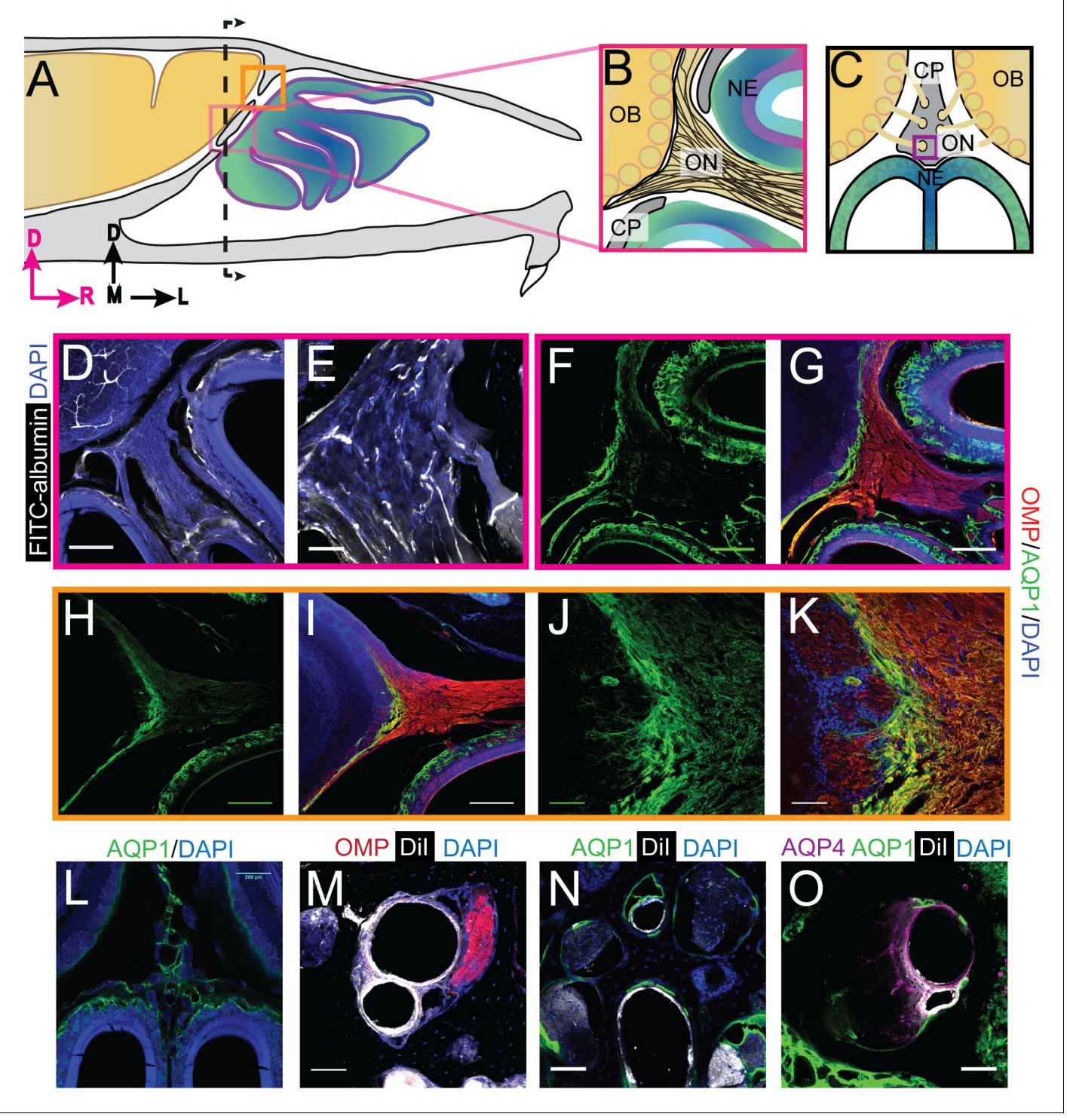

**Figure 2.** Aquaporin 1 and blood vessels are present in the cribriform plate and olfactory nerve junction. For schematics: olfactory nerve (ON), neuroepithelium (NE), cribriform plate (CP), olfactory bulb (OB) and glomeruli (yellow circles). D = dorsal, R = rostral, M = medial, and L = lateral). (A) Schematic of the sagittal plane of the mouse skull and brain showing the relationship of the OB and nerve junction to the CP. (B) Sagittal view of the area within the pink box in (A), depicting OSNs crossing the CP and terminating in the OB and glomeruli. (C) Coronal view of the black dashed line in (A) illustrating the location of the CP relative to the OBs and NE. D) Area depicted by the pink box in (A), showing the presence of FITC-albumin filled blood vessels (white) along the medial ON. (E) Magnified area of (D). (F–O) Immunofluorescent staining, OMP (red), AQP1 (green), AQP4 (purple), DiI (white), and DAPI (blue). (F–G) Area depicted by the pink box in (A), showing the expression of AQP1 (F) at the junction of the OB and medial ON and in the lamina propria of the NE. (H–I) Area depicted by the orange box in (A), showing the expression of AQP1 at the junction of the OB and lateral ON

*Figure 2 continued on next page*

*Figure 2 continued*

and in the lamina propria of the NE. (J–K) Magnified area of (H–I). (L) AQP1 expression, in the area indicated in (C), is present in the lining of the smaller foramina along the midline of the crista galli, on the olfactory nerve layer of the OB, and in the lamina propria of the NE. (M–O) DiI-labeled endothelial cells of blood vessels traversing the smaller foramina of the CP alongside ONs (M) and lined with AQP-1 (N–O) or AQP-4 (O), area indicated by purple box in (C). (D, F–I, L) Scale bars 250 µm. (E, J–K, M–O) Scale bars 50 µm.

DOI: https://doi.org/10.7554/eLife.44278.004

The following figure supplement is available for figure 2:

**Figure supplement 1.** Localization of aquaporins at the olfactory nerve and bulb junction.

DOI: https://doi.org/10.7554/eLife.44278.005

out of the olfactory bulb and subarachnoid space, into the nasal cavity (via OSNs). We observed high levels of AQP1 expression on the olfactory nerve layer down to the glomeruli, as observed previously (*Shields et al., 2010*), and in the lamina propria of the neuroepithelium at both the medial (*Figure 2F–G*) and dorsal (*Figure 2H–K*) junctions of the olfactory bulb and nerve. AQP1 was also observed lining the foramina of the CP along the crista galli (*Figure 2L,N–O*) but did not co-label with Vimentin-positive fibroblasts (*Figure 2* – figure supplement J-K). Based on previous reports (*Au et al., 2002*; *Kumar et al., 2018*) and our own results, we hypothesize that these AQP1-expressing cells might be olfactory ensheathing cells (OECs). High levels of AQP1 expression were not seen in any other brain area, with the exception of the choroid plexus (*Johansson et al., 2005*) strongly suggesting the interface between the olfactory nerve and bulb is a brain region with a high fluid flux flowing through it.

We then examined the expression of AQP4, which is enriched around astrocyte endfeet in the cortex (*Nielsen et al., 1997*) and is hypothesized to play a role in transport of fluid through the glymphatic system (*Iliff et al., 2012*) (though this is controversial [*Smith et al., 2017*]). We observed expression of AQP4 in GFAP-positive astrocytes in the cortex (*Figure 2* – figure supplement L-M) and within the olfactory nerve and neuroepithelium (*Figure 2—figure supplement 1F–G*). The localization of other aquaporins, AQP2, AQP3, and AQP5, were also examined at the olfactory bulb and nerve junction, but their expression was limited to the neuroepithelium and CP (*Figure 2—figure supplement 1D–E,H–I*). The highly enriched expression of aquaporins observed in the tissues at the junctional area of the olfactory nerve and bulb could provide a low-resistance pathway for fluid to pass from the olfactory bulb to the olfactory nerves, out of the cranium, and ultimately into the nasal cavity.

## Lymphatic vessels traverse the cribriform plate

Lymphatic vessels play a role in moving fluid throughout the body and are found in the meninges (*Aspelund et al., 2015*; *Louveau et al., 2015*) and nasal epithelium (*Johnston et al., 2004*). Lymphatic vessels crossing the CP could play a role in transporting fluid from the cranial cavity to the nasal cavity. We used LYVE1-GFP (lymphatic vessel endothelial hyaluronan receptor 1) or LYVE1-tdTomato mice (see Materials and methods) to visualize lymphatic vessels. To verify that the LYVE1-GFP or LYVE1-tdTomato transgene was not expressed in vascular endothelial cells, the mice were perfused using either DiI or FITC-albumin, respectively, and the cervical lymph nodes were examined to detect cross-labeling (*Figure 3—figure supplement 1A–B*). Expression of *LYVE1* was observed in the dural lymphatics, while minimal cross-labeling with fluorescent dye (rhodamine) conjugated to phalloidin (which will preferentially label F-actin of arteries [*Wulf et al., 1979*]) was observed (*Figure 3—figure supplement 1C–D*). Surprisingly, we observed blood vessel (identified by FITC-albumin fills) endothelial cells that expressed *LYVE1* throughout the brain (*Figure 3—figure supplement 1F–G*). We also observed non-lymphatic cells of an unknown type in the frontal cortex expressing *LYVE1* (*Figure 3—figure supplement 2*) that were GFAP, NeuN, Oligo1, and CD164 negative (*Figure 3—figure supplement 2B–E,H–M*). However, they were occasionally MRC1 (mannose receptor 1) positive (*Figure 3—figure supplement 2F–G*), CD31 (PE-CAM) positive (*Figure 3—figure supplement 2N–O*), or expressed in the smooth muscle of arteries and capillaries (*Figure 3—figure supplement 2P–Q*). Interestingly, we found a few LYVE1$^+$ vessels (both DiI and FITC-albumin negative) adjacent to OSNs in the nasal cavity (*Figure 3D–G*), lining holes in the lamina propria of the neuroepithelium, and traversing the CP (*Figure 3H–K*). Although *LYVE1* is not expressed

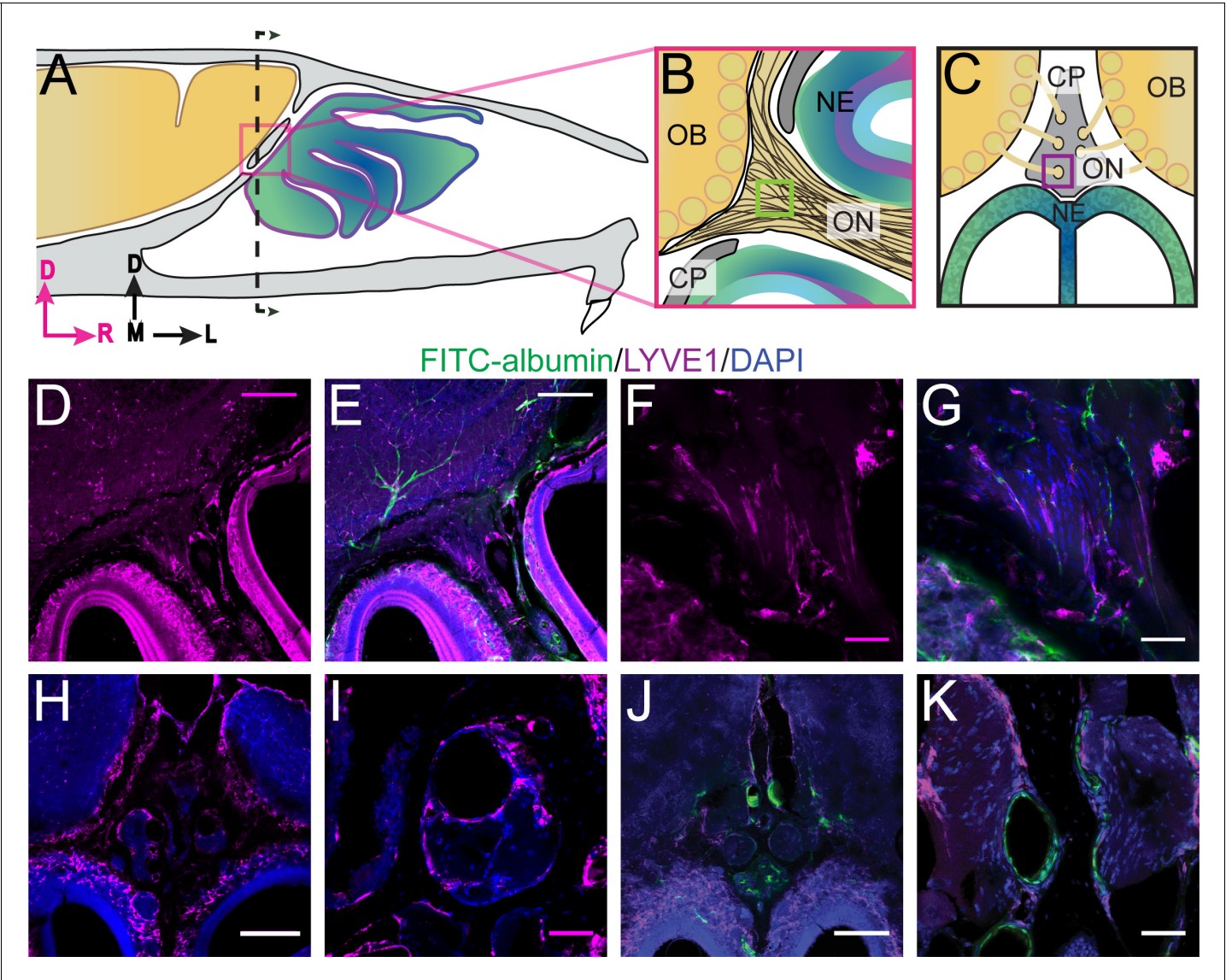

**Figure 3.** Localization of lymphatic vessels at the cribriform plate and olfactory bulb junction. For schematics: olfactory nerve (ON), neuroepithelium (NE), cribriform plate (CP), olfactory bulb (OB) and glomeruli (yellow circles). D = dorsal, R = rostral, M = medial, and L = lateral. (A) Schematic of the sagittal plane of the mouse skull and brain showing the relationship of the OB and nerve junction to the CP. (B) Sagittal view of the area within the pink box in (A), depicting OSNs crossing the CP and terminating in the OB and glomeruli. (C) Coronal view of the black dashed line in (A) illustrating the location of the CP relative to the OBs and NE. (D–K) Immunofluorescent staining: LYVE1 (magenta), FITC-albumin (green), and DAPI (blue) in LYVE1-tdtomato mice. (D–E) Localization of LYVE1[+] vessels along the medial olfactory nerve, sagittal area indicated in (B), depicting LYVE1[+]/FITC-albumin negative vessels (putative lymphatic vessels) running parallel to the olfactory nerve. (F–G) Area indicated by green box in (B), magnified area of (D–E). (H) Localization of LYVE1[+] vessels in the nasal epithelium and traversing the CP, coronal area indicated in (C). (I) Area indicated by purple box in (C), magnified area of (H), depicting LYVE1[+] vessels traversing foramina of the CP. (J) Localization of LYVE1[+] vessels and blood vessels (labeled with FITC-albumin) in the nasal epithelium and traversing the CP, coronal area indicated in (C). (K) Area indicated by purple box in (C), magnified area of (H), depicting only blood vessels (labeled with FITC-albumin) traversing foramina of the CP. (D–E, H, J) Scale bars 250 μm. (F–G, I, K) Scale bars 50 μm.
DOI: https://doi.org/10.7554/eLife.44278.006

The following figure supplements are available for figure 3:

**Figure supplement 1.** Lymphatic vessels in lymph nodes and dura express LYVE1, as do non-lymphatic vessels and cells in the brain.
DOI: https://doi.org/10.7554/eLife.44278.007

**Figure supplement 2.** LYVE1 expression in the murine brain is not cell-type specific.
DOI: https://doi.org/10.7554/eLife.44278.008

exclusively in lymphatic vessels, these results demonstrate that LYVE1[+] lymphatic vessels traverse the CP and could facilitate the movement of fluid from the brain compartment to the nasal compartment.

## Visualization of CSF flow through the cribriform plate

To visualize CSF flow in the regions downstream from the cisterna magna, we injected the high-contrast Evans blue (EB) (*Wang et al., 2015*; *Maloveska et al., 2018*; *Harrell et al., 2008*; *Leinenga et al., 2016*) dye into the cisterna magna (*Figure 4A*). When injected into the blood stream, EB binds to albumin in the plasma making it a high-molecular-weight tracer. However, protein concentrations are vastly lower in the CSF (*Redzic et al., 2005*), so most EB will be unattached to proteins and function as a low molecular weight tracer, freely transported by fluid convection (See Materials and methods for calculations). Injected heads were then decalcified and cut along the midline for imaging as described above (*Figure 4B,D*). EB was not observed within the sagittal sinus or the parenchyma, consistent with EB not crossing the glia limitans. Dye was restricted to the subarachnoid space around the brain and spinal column (*Figure 4B–C*) and was also clearly visible in the nasal epithelium and the deep cervical lymph nodes (*Figure 4E*). To quantify dye movement down the spinal columns, spinal columns were decalcified and cleared using SeeDB (*Ke et al., 2013*; *Hartmann et al., 2015*) (*Figure 4C*). Cardiac perfusion had no qualitative effect on EB movement, as EB was present in the nasal epithelia even without perfusion. The observed pathways of CSF (EB dye) flow and drainage through the CP, into the deep cervical lymph nodes, and along the spinal column, are consistent with previous work using radiolabeled tracers, latex fills, and other tracers (*Bradbury et al., 1981*; *Johnston et al., 2004*; *Cserr et al., 1992b*; *Pizzo et al., 2018*; *Hubbs et al., 2012*). We also injected 3kDa-FITCDextran into the cisterna magna to visualize CSF flow and drainage (*Figure 5B–C*). To determine the anatomic route of the dye, 3kDa-FITCDextran was injected directly into the olfactory bulb. The tracer appeared to be travelling along perivascular space vessels found in the olfactory nerve bundle and the extracellular space. No co-localization of CSF tracers with OMP or β3-tubulin (both label olfactory sensory axons) were observed in the olfactory nerves (*Figure 5—figure supplement 1*). To determine the role of LYVE1[+] vessels in CSF clearance in the nasal cavity, LYVE1-tdTomato (Ai14) mice were injected with 3kDa-FITCDextran in the cisterna magna (*Figure 5D–I*). 3kDa-FITCDextran was observed draining into the nasal cavity along the olfactory nerve (*Figure 5D–F*), and co-localizing with LYVE1[+] vessels along the olfactory nerve (*Figure 5G–I*), although the dye did not appear to be concentrated in the LYVE1[+] vessels. These results show that CSF exits the cranial cavity through the CP and is absorbed by nasal LYVE1[+] (putatively lymphatic) vessels.

## Chemical ablation of olfactory sensory nerves

We then asked if ablation of OSNs, which removes the low-resistance pathway for fluid flow through the CP, might also block movement of CSF through the CP. Previous work has shown that intranasal treatment of $ZnSO_4$ causes the rapid death of OSNs, and persists for months (*Burd, 1993*; *McBride et al., 2003*; *Stewart et al., 1983*). To chemically ablate OSNs, mice were briefly anaesthetized with isoflurane, and an intranasal solution of $ZnSO_4$ (10% in sterile $H_2O$) or vehicle (sterile $H_2O$) was administered to the left nare. Because a loss of the olfactory nerve layer and overall reduction in the size of the olfactory bulb indicates successful ablation of OSNs by $ZnSO_4$ (*Burd, 1993*), we measured the size of the olfactory bulb 2, 10, and 30 days after zinc treatment. A significant decrease in olfactory bulb area was observed compared to vehicle controls 10 days (t(12) = 6.46, p≤0.001, n = 14 (7 each group), ttest2) and 30 days (t(12) = 6.712, p≤0.001, n = 7 for each group, ttest2) after treatment (*Figure 6—figure supplement 1A*). Mice treated with $ZnSO_4$ exhibited weight loss initially after treatment, but then began gaining weight at a normal rate starting one week later (*Figure 6A*). Running wheels (Med Associates Inc, Fairfax, VT) were placed in the cages of individually-housed mice (*Figure 6B*) to quantify activity levels before and after treatment. $ZnSO_4$ treatment had minimal effects on running behavior (*Figure 6C–D*), compared to vehicle controls. These results show that chemical ablation of OSNs with $ZnSO_4$ had minimal effects on the overall health and behavior of the mice.

The olfactory marker protein (OMP) antibody was utilized to histologically examine OSN damage (*Figure 6F–I*). Two days after $ZnSO_4$ treatment (*Figure 6G*), we observed the beginnings of

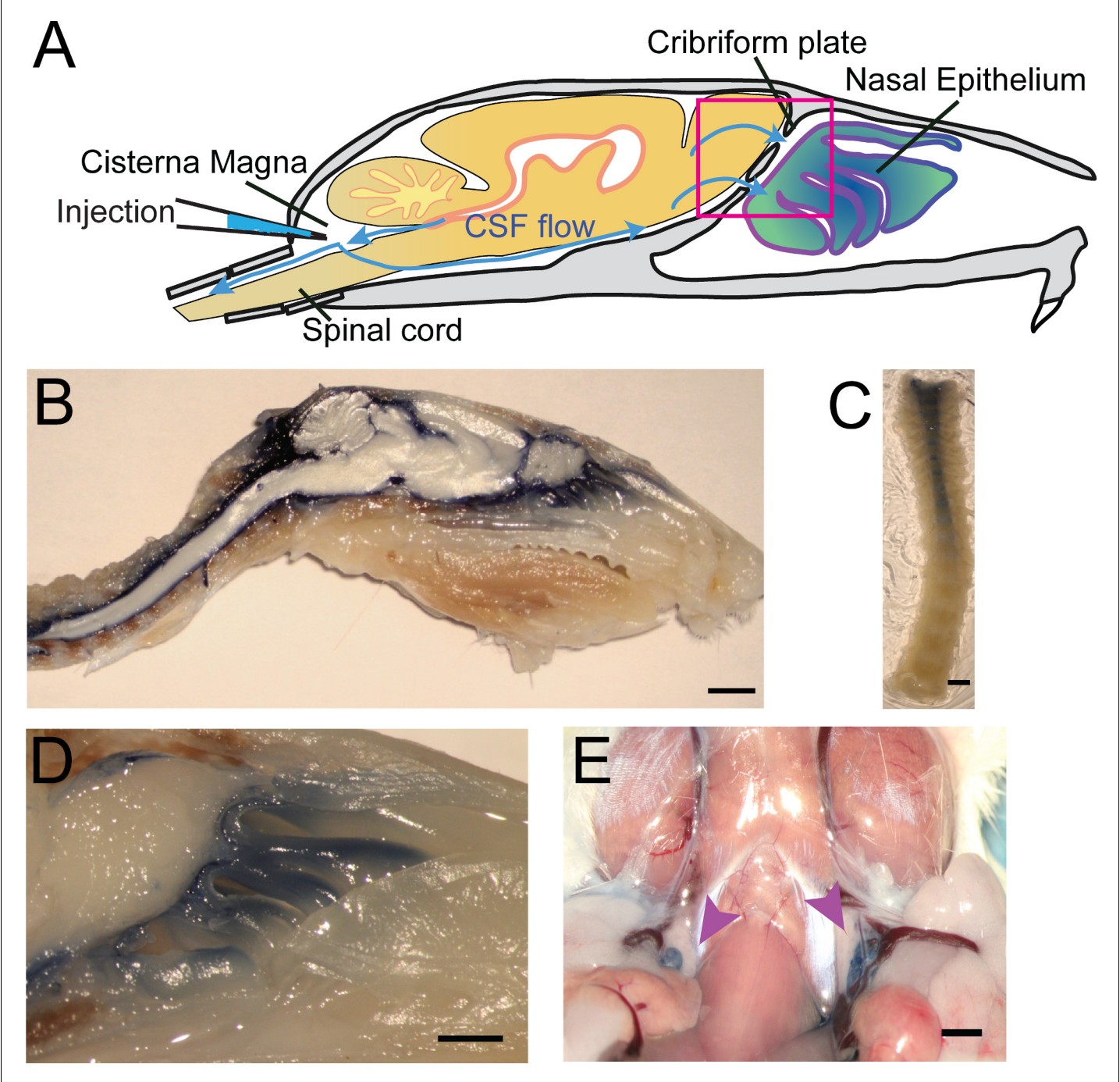

**Figure 4.** Visualization of CSF flow into the nasal cavity and spinal column by injection of Evans blue dye into the cisterna magna. (**A**) Schematic of the sagittal plane of the mouse skull and brain depicting the flow of CSF (blue arrows) and location of EB injection into the cisterna magna. (**B**) Sagittal, midline cut of a decalcified skull and spinal column after a cisterna magna EB injection. Scale bar 3 mm. (**C**) A decalcified and SeeDB-cleared spinal column (cut between C2 and C3, with C3 at the top of the picture) after a cisterna magna EB injection. Scale bar 2 mm. **D**) Area depicted by pink box in (**A**), showing drainage of EB dye across the CP into the nasal cavity. Scale bar 1 mm. (**E**) Localization of EB dye in the deep cervical lymph nodes (purple arrows) after a cisterna magna EB injection. Scale bar 1 mm.

DOI: https://doi.org/10.7554/eLife.44278.009

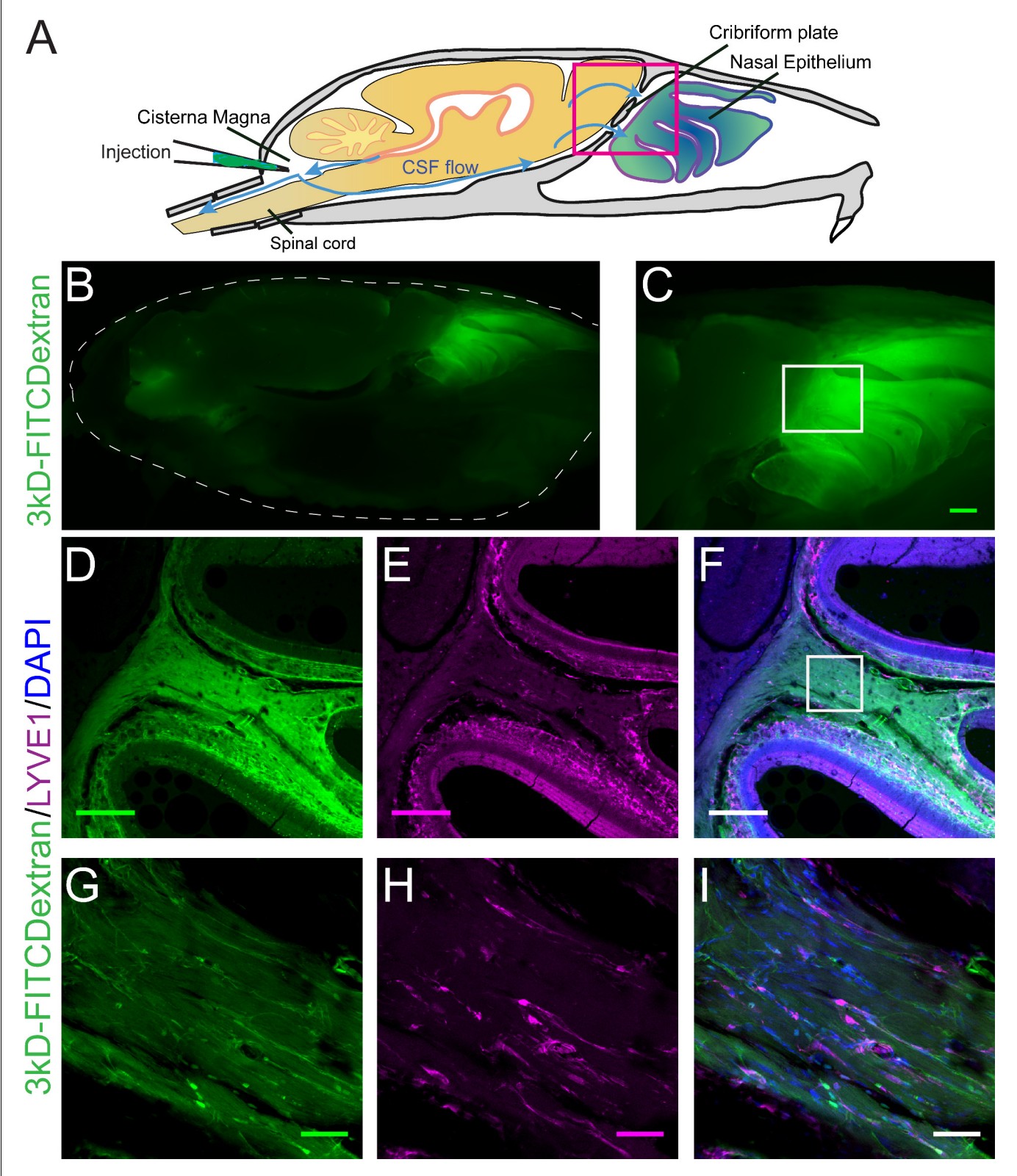

**Figure 5.** Visualization of CSF flow into the nasal cavity by injection of 3kDa-FITCDextran into the cisterna magna. (**A**) Schematic of the sagittal plane of the mouse skull and brain depicting the flow of CSF (blue arrows) and location of 3kDa-FITCDextran injection into the cisterna magna. (**B–I**) Fluorescent images of a mouse skull and brain 35 min after a 3kDa-FITCDextran (green) cisterna magna injection. (**B**) Skull outlined by white dashed lines. 3kDa-FITCDextran is observed in the cisterna magna and the nasal cavity. (**C**) Area indicated by pink box in (**A**). FITC 3kD-Dextran is observed in the nasal

*Figure 5 continued on next page*

*Figure 5 continued*

cavity. Scale bar 500 μm. (**D–I**) Fluorescent images of a mouse skull and brain 35 min after a 3kDa-FITCDextran (green) cisterna magna injection into a LYVE1-tdTomato (Ai14) mouse. LYVE1 (magenta) and DAPI (blue). (**D–F**) Area indicated by white box in (**C**). 3kDa-FITCDextran is observed draining into the nasal cavity along the olfactory nerve. LYVE1$^+$ vessels are observed running parallel to the olfactory nerve. Scale bar 250 μm (**G-I**) Area indicated by white box in (**F**). 3kD-FITCDextran is observed moving along and through LYVE1$^+$ vessels. Scale bar 50 μm.

DOI: https://doi.org/10.7554/eLife.44278.010

The following figure supplement is available for figure 5:

**Figure supplement 1.** Anatomic route of olfactory bulb injected 3kDa-FITCDecxtran dye.

DOI: https://doi.org/10.7554/eLife.44278.011

degeneration of the olfactory nerves. Ten days after ZnSO$_4$ treatment (*Figure 6H*), we observed a profound loss of OSN axons, and 30 days after treatment, a complete loss of OSNs traversing the CP from the nasal epithelium was observed (*Figure 6I*). We quantified the OMP fluorescent signal intensity in the ZnSO$_4$-treated and vehicle mice (*Figure 6N*, *Figure 6—figure supplement 1B*). A significant decrease in OMP signal was observed in the treated animals compared to the vehicle controls (F(3, 32)=259.5, p≤0.001, n = 14 for vehicle, n = 5 for 2 day, n = 8 for 10 day, and n = 9 for 30 day, one-way ANOVA). These results show that OSNs have largely degenerated 10 days after treatment and are absent a month after intranasal ZnSO$_4$ treatment. As a check for any strain-dependent differences, ZnSO$_4$ treatment was also performed in C57BL/6J mice. Both C57BL/6J and Swiss Webster mice were treated with ZnSO$_4$ (or vehicle control) and histologically examined using Nissl-thionin staining 30 days after treatment (*Figure 6—figure supplement 1C–F*), showing similar effects.

We then assayed mRNA levels in the choroid plexus and GFAP expression in the cortex to look for immune responses to ZnSO$_4$ treatment. AQP1, expressed by epithelial cells in the choroid plexus, is involved in CSF production (*Oshio et al., 2005*). Klotho is a protein involved in suppressing aging and is significantly elevated in the CSF of Alzheimer's disease patients (*Kuro-o et al., 1997*; *Semba et al., 2014*). The NF-κβ pathway is involved in the choroid plexus acute-phase response to peripheral inflammation (*Marques et al., 2009*), and RelA is required for NF-κβ activation (*Chen and Greene, 2004*). GFAP is an astrocytic marker and is upregulated as part of an inflammatory response (*Sofroniew, 2009*). However, we found no appreciable changes in mRNA levels or GFAP expression (*Figure 6—figure supplement 2*) 2 or 10 days after ZnSO$_4$ treatment, indicating there is no inflammatory response in the cranial cavity following treatment.

We then asked if nasal lymphatic vessels were damaged by the ZnSO$_4$ treatment. If lymphatic vessels are a conduit for fluid flow, then damage by ZnSO$_4$ would impede outflow. To detect any changes in lymphatic vessel number or morphology, we histologically examined LYVE1-GFP (Ai6) mice 2, 10, and 30 days after treatment, compared to vehicle controls (*Figure 5J–M*). We found no significant change in *LYVE1* expression or localization 2 days after ZnSO$_4$ treatment in LYVE1-GFP mice (*Figure 5K*), however, a change in localization at 10 and 30 days after treatment was observed (*Figure 5L–M*). These results show that ZnSO$_4$ treatment ablates OSNs while not affecting neighboring lymphatic vessel density in the CP area.

## Ablation of OSNs decreases CSF outflow

Because the space between the OSN axons provides a conduit for the outflow of CSF, removing these axons should decrease outflow of CSF. To visualize any disruption of CSF drainage through the CP after OSN ablation, we injected EB into the cisterna magna (*Figure 7A,C,D*, and *Figure 7—figure supplement 1A*) of mice treated with ZnSO$_4$ and vehicle controls (*Figure 7B*). Then the mouse was perfused 10 min after the end of the dye infusion. We quantified the average intensity of EB along the rostral-caudal axis within a region of interest (ROI) that encompassed the olfactory bulb and nasal epithelia (see Materials and methods) (*Figure 7B–H*). Decreased flow of CSF through the CP would appear as a caudal-shift (leftward-shift) in the dye intensity curve. We plotted the average intensity of dye along the rostral-caudal axis for vehicle and ZnSO$_4$-treated animals, for both the ipsilateral (*Figure 7H*, *Figure 7—figure supplement 1B–E*) and the contralateral (*Figure 7—figure supplement 1F–J*) sides of treatment. We quantified dye movement using the center of mass along the anterior-posterior axis (*Figure 7I* and *Figure 7—figure supplement 1K*), the peak location along the anterior-posterior axis (*Figure 7J* and *Figure 7—figure supplement 1L*), and the peak value (*Figure 7K* and *Figure 7—figure supplement 1M*) of the dye intensity. We found that 10 and 30

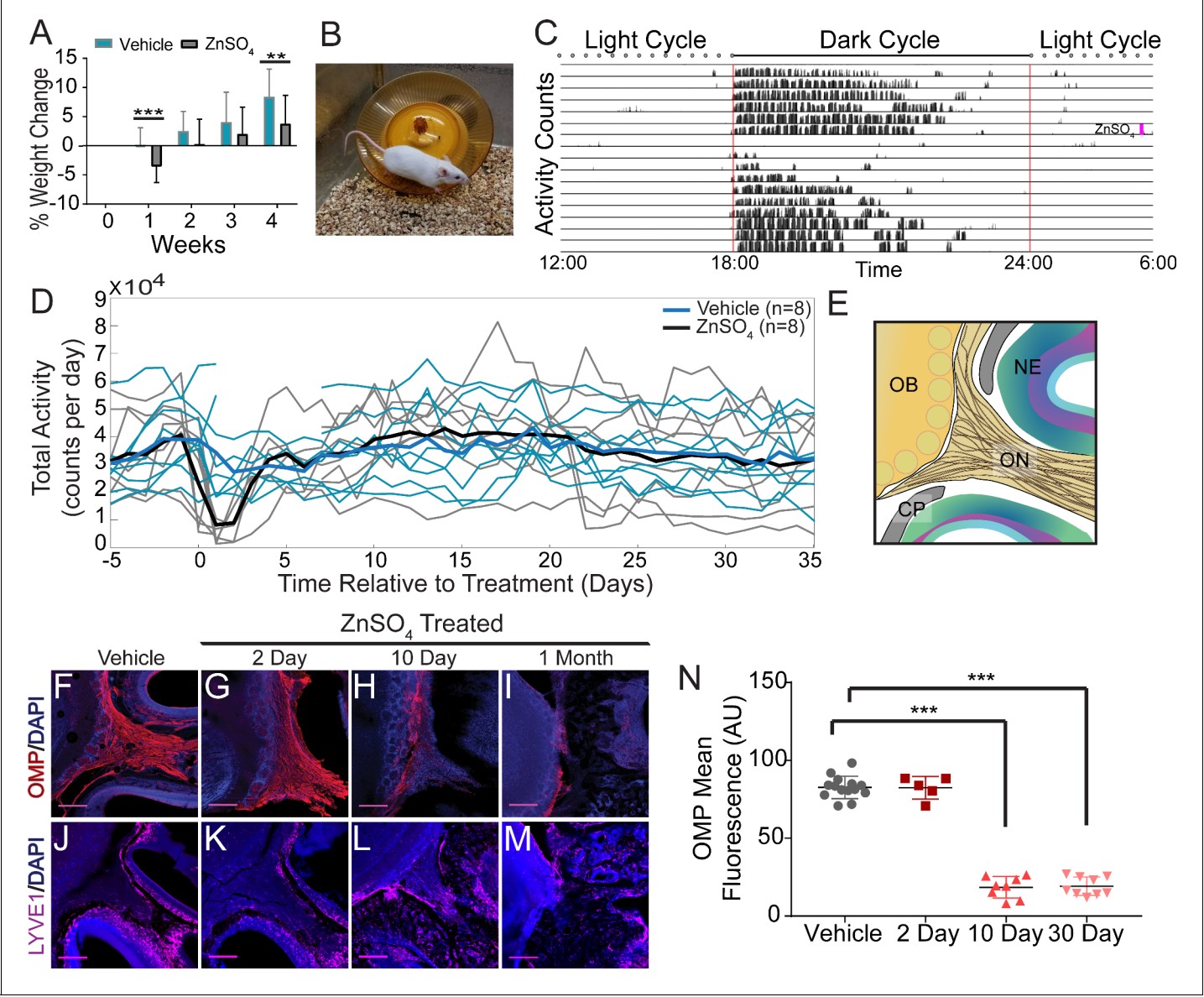

**Figure 6.** Effects of ZnSO$_4$ treatment on behavior and olfactory sensory nerve axons. (A) Percent weight loss of vehicle (blue) and ZnSO$_4$ treated (black) mice after treatment. There was a significant difference in weight of ZnSO$_4$-treated mice compared to control mice 1 week (p=0.001, n = 17 for vehicle and n = 25 for treated, ttest2) and 4 weeks (p=0.004, n = 17 for vehicle and n = 25 for treated, ttest2) after treatment. Mean ± standard deviation plotted. **p≤0.01 ***p≤0.001 (B) Photo of a Swiss Webster mouse on the running wheel. (C) Actogram of the activity of a single mouse before and after ZnSO$_4$ treatment. Treatment time indicated by pink line. (D) Individual counts of total wheel activity per day (vehicle = blue, ZnSO$_4$treated = gray), bold line is the mean of each group. No significant difference observed between groups (p=0.5189, KSTAT = 0.3750, n = 8 for each group, K-S test). (E) Schematic: sagittal plane depicting OSNs crossing the CP and synapsing onto the OB and glomeruli (yellow circles). (F–M) Immunofluorescent staining of area depicted in (E), OMP (red), LYVE1 (magenta) and DAPI (blue). Scale bar 250 μm. (F, J) Vehicle Control. (G, K) Two days after ZnSO$_4$ treatment. (H, L) Ten days after ZnSO$_4$ treatment. (I, M) Thirty days after ZnSO$_4$ treatment. Swiss Webster mice were used for (F–I) and LYVE1-GFP (Ai6) mice were used for (J–M). (N) Quantification of OMP signal fluorescence for vehicle (n = 14) mice compared to 2 (t(8) = 0.1429, p=0.8898, n = 5, post-hoc ttest2), 10 (t(13) = 15.71, p≤0.001, n = 8, post-hoc ttest2) and 30 (t(13) = 25.39, p≤0.001, n = 9, post-hoc ttest2) days after ZnSO$_4$ treatment. Mean ± standard deviation plotted. Circles, triangles, and squares represent means of individual animals. ***p≤0.001.

DOI: https://doi.org/10.7554/eLife.44278.012

The following figure supplements are available for figure 6:

**Figure supplement 1.** ZnSO$_4$-treatment causes olfactory bulb degeneration.

DOI: https://doi.org/10.7554/eLife.44278.013

**Figure supplement 2.** Genetic and histological assays show no inflammatory response in the brain or choroid plexus to ZnSO$_4$ treatment.

*Figure 6 continued on next page*

*Figure 6 continued*

DOI: https://doi.org/10.7554/eLife.44278.014

days after treatment, the peak location and center of mass of the dye curve was shifted caudally in ZnSO$_4$-treated mice, relative to vehicle controls, indicating that CSF outflow through the CP was decreased by OSN ablation. For the side ipsilateral to treatment, we saw significant interactions between the effects of time after treatment and the treatment type (ZnSO$_4$ or vehicle) on the center of mass location (*Table 1*). We also saw significant effects on the center of mass and the anterior-posterior peak location by both the duration of treatment and the type of treatment (*Table 1*). Next, to compare the amount of EB collected in the deep cervical lymph nodes after a cisterna magna injection between zinc- and vehicle-treated mice, the volume of EB injected (2 μL to 4.5 μL) and subsequently the duration of infusion was increased to 45 min. No difference in EB collected in the deep cervical lymph nodes was observed between vehicle- (*Figure 8B*) and zinc-treated (*Figure 8C*) mice 30 days after treatment. Interestingly, the decrease in CSF outflow through the CP in zinc-treated mice, compared to vehicle mice, was still observed 30 days after treatment (*Figure 8D–E*), despite the increases in infusion volume and duration. While a significant caudal shift was observed in the center of mass of the EB dye peak, no difference was observed in the dye peak location (*Figure 8F–G*) or the distance dye traveled along the spinal column (*Figure 8H*). To confirm that this decrease of outflow was not due to some unique aspect of EB, we repeated these experiments by injecting 5 μL of 10% 3kDa-FITCDextran into the cisterna magna. For 3kDa-FITCDextran infusions, an infusion rate of 0.2 μL/min, was used and the animals were euthanized 10 min after the end of the infusion. A significant decrease in outflow of 3kDa-FITCDextran into the nasal cavity in ZnSO$_4$-treated mice, relative to vehicle controls, was also observed (*Figure 9*). Thus, ablating OSNs blocks CSF outflow through the CP, and this blockage persists for a least a month.

## OSN ablation does not affect ICP

The production and outflow of CSF is dynamically balanced (*Marmarou et al., 1975*) in such a way that an increase in outflow resistance will drive a rise in ICP (*Kosteljanetz, 1987*; *Jones, 1985*), unless there is a compensatory reduction of CSF production or changes in the resistance of other outflow pathways. To test if the disruption of the nasal CSF outflow pathway causes a rise in ICP, we measured the ICP from awake, head-fixed mice (*Gao and Drew, 2016*; *Gao et al., 2017*). We implanted a titanium headbar and habituated the mice to head fixation on a spherical treadmill (*Gao and Drew, 2014*), allowing them to voluntarily locomote. Locomotion is a natural behavior that causes vasodilation across the cortex (*Huo et al., 2014*), which will increase ICP (*Dickinson et al., 2016*). As a positive control, we also measured ICP in mice with cisterna magna injection of aCSF or kaolin (*Figure 10A*). Injections of kaolin into the cisterna magna are known to produce elevations in ICP, and are used to generate an animal model of hydrocephalus (*Bloch et al., 2006*). After mice were habituated to head fixation, kaolin or aCSF cisterna magna injections were performed. For ICP measurements, a small craniotomy was performed under isoflurane anesthesia and an ICP sensor (Millar SPR-1000) was placed ~1 mm into the cortex. Mice were allowed to wake up from anesthesia on a spherical treadmill, where ICP and treadmill motion were recorded. Data collection started at least 1 hr after the cessation of anesthesia to avoid the effects of anesthetics on ICP and other physiological processes (*Gao and Drew, 2016*; *Gao et al., 2017*; *Shirey et al., 2015*). We then measured ICP in mice intranasally treated with vehicle or ZnSO$_4$, utilizing the same procedure for measuring ICP of the aCSF and kaolin injected mice (*Figure 10A*).

We found that 2 days after kaolin injections into the cisterna magna ICP significantly increased during locomotion (*Table 2*) compared to aCSF controls (*Figure 10B–C*), but not during rest. No statistically significant difference was observed in ZnSO$_4$-treated mice at rest or during locomotion as compared to the vehicle controls (*Figure 10B–C*) measured 10, 30, and 60 days after treatment (*Table 2*). Additionally, there was no difference between the groups' ICP immediately after isoflurane cessation and before the animal regained consciousness (*Figure 10D*, *Table 2*). Despite the blockage of the nasal CSF outflow pathway, ICP did not increase. This suggests that an alternate outflow route is taken and/or CSF production is decreased.

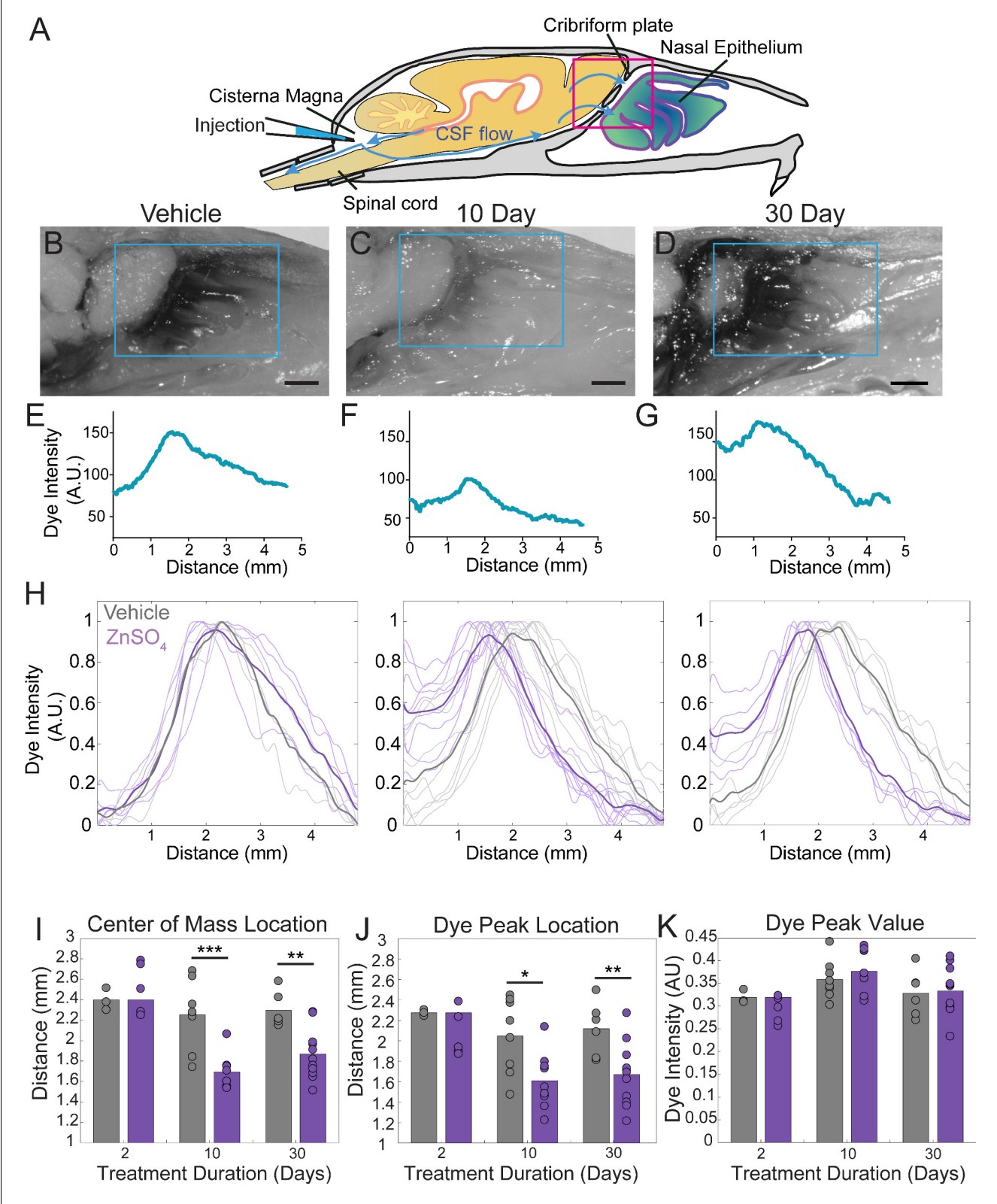

**Figure 7.** ZnSO$_4$ treatment decreases CSF outflow through the cribriform plate ipsilateral to treatment. (**A**) Sagittal view schematic depicting direction of CSF flow and location of a cisterna magna injection of Evans blue. (**B–D**) Sagittal view of decalcified and cut skull for imaging after Evans blue injection, area indicated by pink box in (**A**). ROI indicated by blue box. Scale bar 1 mm. (**B**) Vehicle control. (**C**) 10 days after ZnSO$_4$ treatment. (**D**) 30 days after ZnSO$_4$ treatment. (**E–G**) Plot of the intensity of the Evans blue dye as a function rostral-caudal distance. (**E**) Vehicle control, ROI of area

*Figure 7 continued on next page*

*Figure 7 continued*

measured indicated by blue box in (B). (F) 10 day ZnSO$_4$ treated, ROI of area measured indicated by blue box in (C). (G) 30 days after ZnSO$_4$ treatment, ROI of area measured indicated by blue box in (D). (H) Comparison of all dye curves of all vehicle- (gray) and ZnSO$_4$-treated (purple) groups for each time point after treatment: 2 day, n = 3 vehicle and n = 5 treated. 10 day, n = 5 for each group. 30 day, n = 6 vehicle and n = 11 for treated. Means plotted as bold lines. (I–K) Mean is plotted as height of bar. Circles represent individual animals. (I) Mean of the center of mass location of the dye curve plotted for vehicle- and ZnSO$_4$-treated animals 2 (t(6) = 0.7489, p=0.4822, n = 3 vehicle and n = 5 treated, post-hoc ttest2), 10 (t(15) = −4.4921, p=0.0004, n = 5 for each group, post-hoc ttest2), and 30 ((t(15) = −3.7539, p=0.0019, n = 6 vehicle and n = 11 for treated, post-hoc ttest2) days after treatment. J) Mean dye peak location plotted for vehicle and ZnSO$_4$-treated animals 2 (t(6) = −1.5082, p=0.1822, n = 3 vehicle and n = 5 treated, post-hoc ttest2), 10 (t(15) = −2.8064, p=0.0133, n = 5 for each group, post-hoc ttest2), and 30 (t(15) = −3.0014, p=0.0089, n = 6 vehicle and n = 11 for treated, post-hoc ttest2) days after treatment. (K) Maximum dye peak value plotted for vehicle and ZnSO$_4$-treated animals 2 (t(6) = −1.4187, p=0.2058, n = 3 vehicle and n = 5 treated, post-hoc ttest2), 10 (t(15) = 0.7982, p=0.4372, n = 5 for each group, post-hoc ttest2), and 30 (t(15) = 0.2052, p=0.8402, n = 6 vehicle and n = 11 for treated, post-hoc ttest2) days after treatment. *p≤0.05 **p≤0.01. ***p≤0.001.

DOI: https://doi.org/10.7554/eLife.44278.015

The following figure supplement is available for figure 7:

**Figure supplement 1.** ZnSO$_4$ treatment impacts CSF outflow on the side contralateral to treatment.

DOI: https://doi.org/10.7554/eLife.44278.016

## Decreased CSF movement down the spinal column following OSN ablation

A block in CSF drainage accompanied by no change in ICP implies either a compensatory *decrease* in CSF production or an increase in outflow along other pathways, such as the spinal column. To distinguish between these two hypotheses, we examined the spinal columns after a cisterna magna injection of EB into mice treated with intranasal ZnSO$_4$ or vehicle. Since the cisterna magna is 'upstream' from all outflow pathways, including the spinal column, if there is a compensatory increase in the outflow of CSF via the spinal pathway, we would expect dye injected into the cisterna

**Table 1.** Two-way ANOVA statistics for all Evans blue cisterna magna injections.
d.o.f. = degrees of freedom. CoM = center of mass, PL = peak location, PV = peak value. All significant p values are bolded (p≤0.05).

**Evans blue drainage**

| F | d.o.f. | P | Side | Parameters | Parameters |
|---|---|---|---|---|---|
| 5.45 | 2 | **0.0086** | IPSI | CoM | Interaction: duration * type |
| 0.49 | 2 | 0.6144 | IPSI | PL | Interaction: duration * type |
| 0.61 | 2 | 0.5494 | IPSI | PV | Interaction: duration * type |
| 11.05 | 2 | **0.0002** | IPSI | CoM | effect: duration |
| 13.16 | 1 | **0.0009** | IPSI | CoM | effect: type |
| 3.63 | 2 | **0.0365** | IPSI | PL | effect: duration |
| 13.72 | 1 | **0.0007** | IPSI | PL | effect: type |
| 5.27 | 2 | **0.0098** | IPSI | PV | effect: duration |
| 0 | 1 | 0.9497 | IPSI | PV | effect: type |
| 1.06 | 2 | 0.3601 | CONTRA | CoM | Interaction: duration * type |
| 2.08 | 2 | 0.1435 | CONTRA | PL | Interaction: duration * type |
| 0.79 | 2 | 0.4645 | CONTRA | PV | Interaction: duration * type |
| 2.93 | 2 | 0.0694 | CONTRA | CoM | effect: duration |
| 3.84 | 1 | 0.0596 | CONTRA | CoM | effect: type |
| 4.19 | 2 | **0.0253** | CONTRA | PL | effect: duration |
| 10.64 | 1 | **0.0028** | CONTRA | PL | effect: type |
| 0.73 | 2 | 0.49 | CONTRA | PV | effect: duration |
| 0.67 | 1 | 0.4188 | CONTRA | PV | effect: type |

DOI: https://doi.org/10.7554/eLife.44278.019

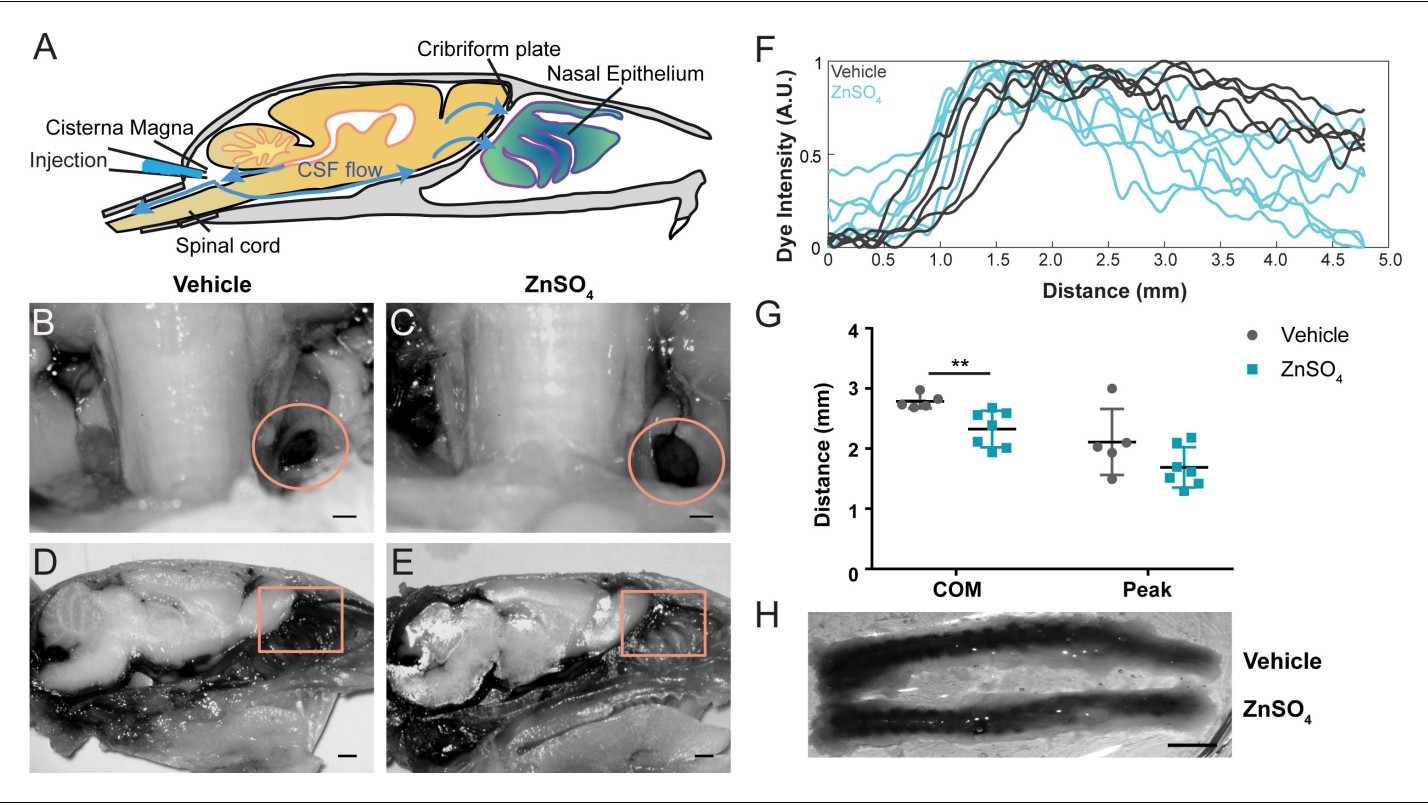

**Figure 8.** Longer-duration dye injections show ZnSO$_4$ treatment blocks CSF flow across the cribriform plate. (A) Sagittal view of the skull depicting direction of CSF flow and location of cisterna magna injection of EB. (B–C) Localization of EB dye in the deep cervical lymph nodes (orange circles) after a cisterna magna EB injection. Scale bar 500 µm. (B) Vehicle control. (C) 30 days after ZnSO$_4$ treatment. (D–E) Sagittal view of decalcified and cut skull of the ipsilateral side for imaging after EB injection in the cisterna magna. ROI indicated by orange box. Scale bar 1 mm. (D) Vehicle control. (E) 30 days after ZnSO$_4$ treatment. (F) Plot of the intensity of the Evans blue dye as a function rostral-caudal distance for vehicle (gray) and ZnSO$_4$-treated (blue) groups 30 days after treatment. ROI indicated by orange box in (D–E). (G) Mean of the center of mass (t(10) = −3.2015, p=0.0095, n = 5 vehicle and n = 7 treated, post-hoc ttest2) and dye peak (t(10) = −1.6604, p=0.1278, n = 5 vehicle and n = 7 treated, post-hoc ttest2) location of the curve plotted for vehicle- and ZnSO$_4$-treated animals 30 days after treatment. (H) Vehicle (top) and 30 days after ZnSO$_4$ treatment (bottom) decalcified and SeeDB cleared spinal columns after cisterna magna EB injection. Scale bar 500 mm.

DOI: https://doi.org/10.7554/eLife.44278.017

magna to move further down the spinal columns of mice with olfactory nerve ablations as compared to vehicle controls. Conversely, if the CSF production is *reduced* to compensate for the loss of the nasal CSF outflow pathway, we would expect to see a reduction in dye movement down the spinal columns. To differentiate these two possibilities, we extracted the spinal columns and cleared them using SeeDB (*Ke et al., 2013*; *Hartmann et al., 2015*) after EB cisterna magna injections. We quantified the distance the dye traveled along the dorsal side of the spinal column for both vehicle and ZnSO$_4$ treated animals, 2, 10, and 30 days after treatment (*Figure 10E–G*). While no significant difference between the vehicle and ZnSO$_4$ treatment groups in the distance of dye traveled 2 days after treatment was observed, a significant decrease in the distance traveled was observed for both 10 and 30 days after treatment (*Figure 10H*). Our results show that in response to the disruption of the nasal outflow pathway there was a decrease in CSF flow down the spinal cord.

## Discussion

Utilizing histology, microCT imaging, physiology, and tracer dye injections, we investigated the anatomical basis of CSF outflow through the CP into the nasal cavity. The foramina will permit the flow of fluid out of the brain through the intercellular space between OSN axons, blood vessels, and LYVE1$^+$ vessels traversing through them. We observed a highly stereotyped pattern in the

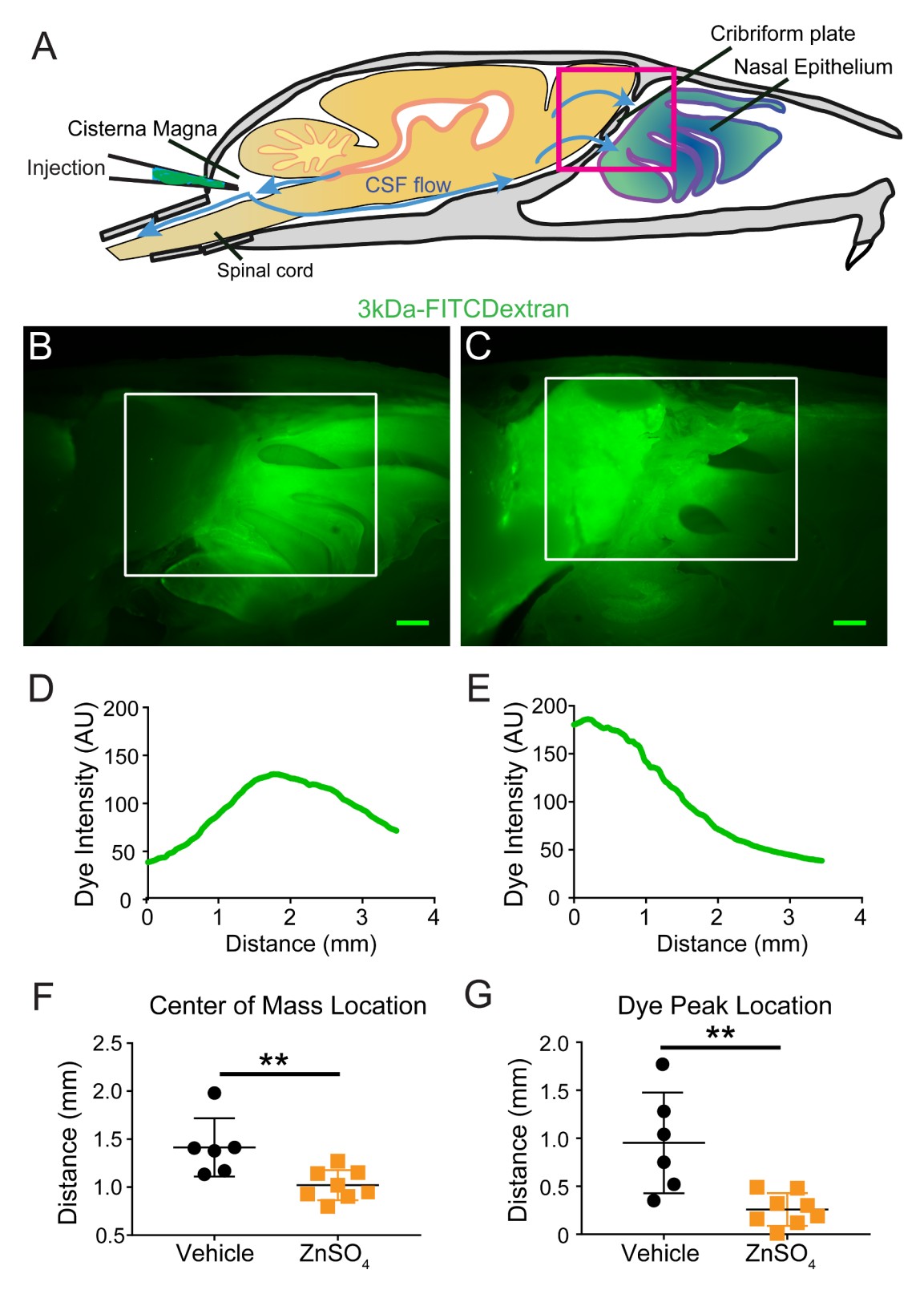

**Figure 9.** Visualization of effects of intranasal ZnSO$_4$ treatment on CSF flow with 3kDa-FITC Dextran. (**A**) Schematic of a sagittal view of the skull, depicting direction of CSF flow and location of cisterna magna injection of 3kDa-FITCDextran. (**B–C**) Sagittal view of decalcified and cut skull for imaging after FITC-dextran injection, area indicated by pink box in (**A**). Scale bar 500 µm. (**B**) Vehicle control. (**C**) 10 day ZnSO$_4$ treated. (**D–E**) Plot of the intensity of the 3kDa-FITC Dextran dye as a function of rostral-caudal distance. (**D**) Vehicle control, ROI of area measured indicated by white box in (**B**).
*Figure 9 continued on next page*

*Figure 9 continued*

(E) 10 days after ZnSO$_4$ treatment, ROI of area measured indicated by white box in (C). (F–G) Mean ± standard deviation plotted. Circles and squares represent means of individual animals. (F) Comparison of the center of mass location of the dye peak between the vehicle (black) and ZnSO$_4$ (orange) treated groups (t(12) = −3.1805, p=0.0079, n = 6 vehicle and n = 8 treated, post-hoc ttest2). (G) Comparison of the location of the dye peak between the vehicle (black) and ZnSO$_4$ (orange) treated groups (t(12) = −3.5384, p=0.0041, n = 6 vehicle and n = 8 treated, post-hoc ttest2). **p≤0.01. Note that the time between cisterna magna injection and sacrifice was different in the FITC-dextran and EB injections (35 vs. 20 min respectively) due to the different molecular weights of the dyes.

DOI: https://doi.org/10.7554/eLife.44278.018

positioning of the major foramina across mice. It is tempting to speculate that the routes of the OSNs are predisposed to go through either the dorsal or ventral foramina depending on their receptor group and target glomeruli (*Buck and Axel, 1991*; *Feinstein and Mombaerts, 2004*; *Rodriguez-Gil et al., 2010*). We found that aquaporin-1 (AQP1) was localized to the olfactory bulb and nerve junction and lining the foramina of the CP, instead of aquaporin-4 (AQP4), which is the dominant water channel in the CNS (*Nagelhus and Ottersen, 2013*). We disrupted CSF drainage through the CP (as well as along the spinal column) by ablating the OSNs. Interestingly, instead of a rise in ICP as would be expected due to a blockage of CSF drainage into the nasal cavity, we observed no change in ICP. This could not be explained by an increase in drainage of CSF down the spinal cord pathway, as we observed a similar reduction in CSF flow down this pathway following ZnSO$_4$ treatment. We can think of two non-mutually exclusive possibilities that could explain the lack of ICP rise when an outflow pathway is blocked. The first is that other drainage pathways become engaged and/or reduce their outflow resistance to compensate for the blockage of the cribriform plate outflow. In addition to providing enough drainage to prevent ICP rise, the CSF flow dynamics would be altered by these outflow resistance changes such that the flow down the spinal cord is decreased. One possible location for outflow increase could be the lymphatics at the skull base (*Antila et al., 2017*) or along the other nerve bundles that exit the skull (*Weller et al., 1992*; *Kida et al., 1993*; *Bradbury and Cole, 1980*). A re-routing of CSF flow in a more rostral direction would account for the decreased flow down the spinal cord. Some outflow pathways have thresholds for opening (*Welch and Friedman, 1960*), and these might become engaged following ZnSO$_4$ treatment. The second possibility could be a decrease in the production of CSF. CSF production is not a passive process, but rather one under active neural control (*Lindvall et al., 1978*; *Edvinsson and Lindvall, 1978*; *Tuor et al., 1990*). As ICP needs to be maintained within a healthy range, there are likely homeostatic processes that sense ICP, and cause compensatory increases or decreases in CSF production, similar to other mechanosensory processes in other organs of the body (*Umans and Liberles, 2018*; *Guild et al., 2018*). A potential mechanism mediating this feedback is mechanically sensitive ASIC3 channels, which are expressed in the ventricles (*Jalalvand et al., 2018*), and are sensitive to changes in pressure that are physiologically relevant to ICP regulation. Pathological elevations of ICP due to trauma produce a rise in systemic blood pressure and a decrease in heart rate, known as the Cushing response (*Hoff and Reis, 1970*; *Doba and Reis, 1972*), showing the brain has a way of sensing ICP (*Hoff and Reis, 1970*), and pressure-sensitive brain regions mediating these responses have been identified (*Hoff and Reis, 1970*; *Doba and Reis, 1972*). A similar response may be occurring in the choroid plexus to compensate for the decreased CSF drainage we observe after ZnSO$_4$ treatment, thus, avoiding the rise in ICP that is normally associated with a decrease in CSF drainage (*Lindvall et al., 1978*; *Edvinsson and Lindvall, 1978*). Neither of these two possibilities excludes the other, and from the experiments performed here, we are unable to differentiate between these two possibilities.

Our dye injections were done under anesthesia and the CSF drainage observed may differ in the awake animal. Although these limitations should be considered, since we observed a difference in drainage between the ZnSO$_4$- and vehicle-treated mice, this should still be considered a significant decrease in CSF drainage due to OSN ablation. Although our lymphatic mouse model (Lyve1.Cre. GFP mice crossed with Ai6 or Ai14 reporter lines) exhibited some non-lymphatic specific *LYVE1* expression in vascular endothelial cells in the brain, there were also LYVE1$^+$ vessels along the olfactory nerve that did not co-label with vascular perfusion markers (FITC-albumin fills or DiI staining) and that absorbed dye injected into the cisterna magna. These vessels are putatively lymphatic vessels that transport CSF out of the cranial cavity. Previous work has shown that there is transport of

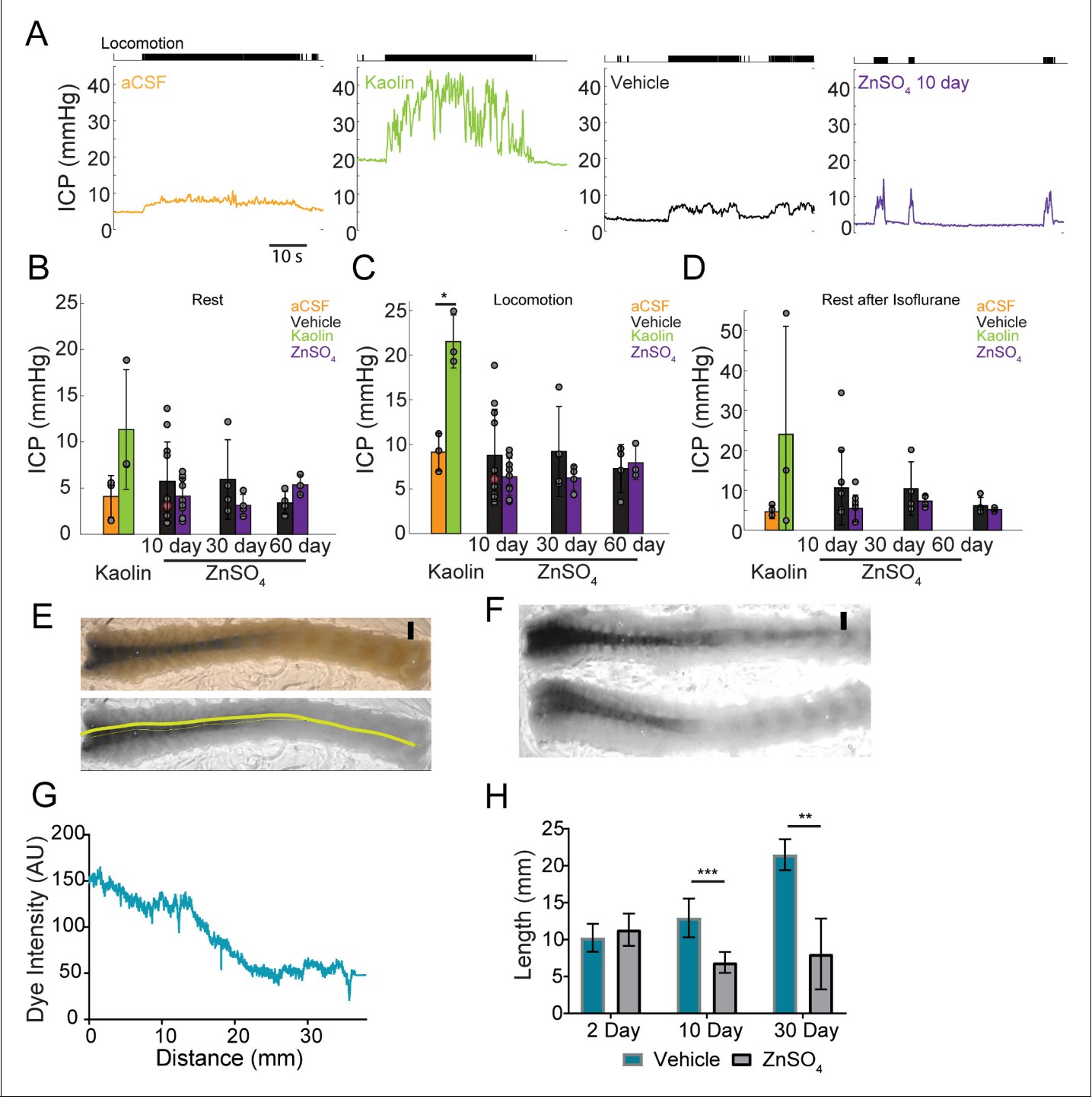

**Figure 10.** Intranasal ZnSO$_4$ treatment does not affect ICP but decreases CSF movement down the spinal column. (A) Left panel, example trace showing intracranial pressure (ICP) change during a locomotion bout in an animal that has been injected with aCSF in the cisterna magna. The ICP becomes elevated slightly over baseline when the animal moves, consistent with previous results (*Gao and Drew, 2016*). The tick marks in the upper panel indicates locomotion events, defined as times where the treadmill acceleration exceeds a threshold (*Huo et al., 2014*). Subsequent panels show ICP change during locomotion in animals injected with Kaolin, vehicle control, or ZnSO$_4$ treatment, respectively. (B–D) Mean ± standard deviation plotted. Circles represent individual animals. (B) Mean of each group of ICP during rest (1 hr after isoflurane) after kaolin injection (Kaolin) and 10, 30, and 60 days after treatment (ZnSO$_4$). The orange shaded circle indicates the same animal showing in (A). (C) Same as (B) but for ICP during locomotion. Comparison of ICP during locomotion for aCSF and kaolin injected mice (t(4) = 5.88, p=0.0042, n = 3 for each group, ttest2). (D) Same as (B) but for stationary (resting) periods in the 2 min immediately after the cessation of isoflurane. (B–D) No statistically significant differences were observed

*Figure 10 continued on next page*

*Figure 10 continued*

between ZnSO₄-treated and control animal at rest or during locomotion for all treatment durations (*Table 2*). (E) Top panel: A decalcified and SeeDB-cleared spinal column after a cisterna magna EB injection. Bottom panel: Red channel only displaying the trace (yellow line) used for quantifying pixel intensity. Scale bar 2 mm. (F) Vehicle (top) and 10 days after ZnSO₄ treatment (bottom) decalcified and SeeDB cleared spinal columns after cisterna magna EB injection. (G) Trace obtained and graphed to display dye intensity of EB along the spinal column (distance) quantified after cisterna magna injections. (H) Comparisons of the distance of dye movement along the spinal column after an EB cisterna magna injection in mice intranasally treated with vehicle or ZnSO₄, 2 (t(5) = 0.6921, p=0.5197, n = 3 vehicle and n = 4 treated, ttest2), 10 (t(12) = 5.069, p=0.00028, n = 8 vehicle and n = 6 treated, ttest2), and 30 days (t(9) = 4.571, p=0.00134, n = 3 vehicle and n = 8 treated, ttest2), after treatment. Mean ± standard deviation plotted. *p≤0.05 **p≤0.01 ***p≤0.001.

DOI: https://doi.org/10.7554/eLife.44278.020

fluid across the CP (*Bradbury et al., 1981*) and acute blockage of the plate (by sealing it with cyano-acrylate glue) blocks transport of radiolabeled albumin (*Bradbury and Westrop, 1983*) establishing the CP as a route of CSF drainage, which then drains into the nasal lymphatics (*Erlich et al., 1986*; *Weller et al., 1992*; *Koh et al., 2006*; *Nagra et al., 2006*; *Ma et al., 2017*; *Kida et al., 1993*; *Walter et al., 2006*; *Ma et al., 2019*; *Ethell, 2014*; *Pizzo et al., 2018*; *Cserr and Patlak, 1992c*; *Papaiconomou et al., 2002*; *Johnston et al., 2005*). Our work elucidates the anatomical basis of CSF transport through the CP and its role in CSF turnover, and demonstrates that damage to the olfactory sensory nerves can impair this pathway.

This work provides a possible mechanism for the observed correlations between neurodegeneration, anosmia, and environmental toxins like air pollution. Although olfactory impairment, or anosmia, occurs during normal aging (*Doty and Kamath, 2014*; *Murphy et al., 2002*), anosmia frequently precedes many neurological disorders (*Talamo et al., 1989*; *Talamo et al., 1991*; *Kovács, 2004*; *Roberts et al., 2016*; *Zou et al., 2016*). Furthermore, studies have shown correlations with the levels of air pollution with olfactory impairment (*Ajmani et al., 2016*; *Doty and Hastings, 2001*; *Adams et al., 2016*; *Imamura and Hasegawa-Ishii, 2016*; *Cheng et al., 2016*) and neurodegenerative disease pathology, such as Alzheimer's disease (*Calderón-Garcidueñas et al., 2004*; *Calderón-Garcidueñas et al., 2012*; *Calderón-Garcidueñas et al., 2016a*; *Calderón-Garcidueñas et al., 2016b*; *Chen et al., 2017a*; *Chen et al., 2017b*), although the mechanism behind these correlations is unclear. Anosmia has many causes, such as smoking and head trauma (*Doty and Hastings, 2001*), and toxin exposure is another source of olfactory deficits. The OSNs are exposed to environmental toxins (such as air pollutants [*Ajmani et al., 2016*]) in the nasal cavity, causing an increased rate of OSN death. A decrease in the number of OSNs may be increasing the resistance to CSF outflow, triggering a downregulation in CSF production to maintain normal ICP and/or a change in other outflow pathways that alter CSF flow dynamics. Reduced CSF turnover

**Table 2.** Mixed model ANOVA statistics for intracranial pressure (ICP) measurements.
For each time point, statistics were calculated for the comparison of the vehicle and treated animal groups. d.o.f. = degrees of freedom.

**Mixed model ANOVA vehicle vs. ZnSO₄ treated**

|        | N(vehicle) | N(treated) | Effect     | F     | d.o.f. | P      |
|--------|------------|------------|------------|-------|--------|--------|
| 10 day | 11         | 10         | treatment  | 1.26  | 37     | 0.268  |
|        |            |            | locomotion | 7.95  |        | 0.0077 |
|        |            |            | isoflurane | 3.38  |        | 0.0827 |
| 30 day | 4          | 4          | treatment  | 3.3   | 13     | 0.0922 |
|        |            |            | locomotion | 7.32  |        | 0.018  |
|        |            |            | isoflurane | 0.57  |        | 0.4786 |
| 60 day | 4          | 3          | treatment  | 1.7   | 11     | 0.2183 |
|        |            |            | locomotion | 11.47 |        | 0.0061 |
|        |            |            | isoflurane | 0.58  |        | 0.4812 |

DOI: https://doi.org/10.7554/eLife.44278.021

may be a contributing factor to the buildup of toxic metabolites and proteins that cause neurodegenerative disorders.

# Materials and methods

## Key resources table

| Reagent type (species) or resource | Designation | Source or reference | Identifiers | Additional information |
|---|---|---|---|---|
| Strain, strain background (*M. musculus*) | Strain: C57BL/6J | Jackson Laboratory | Stock No: 000664; RRID:IMSR_JAX:000664 | |
| Strain, strain background (*M. musculus*) | Strain: LYVE1Cre.GFP | Jackson Laboratory | Stock No: 012601; RRID:IMSR_JAX:012601 | |
| Strain, strain background (*M. musculus*) | Strain: Ai6 (RCL-ZsGreen) | Jackson Laboratory | Stock No: 007906; RRID:IMSR_JAX:007906 | |
| Strain, strain background (*M. musculus*) | Strain: Ai14 (RCL-tdT)-D | Jackson Laboratory | Stock No: 007914; RRID:IMSR_JAX:007914 | |
| Strain, strain background (*M. musculus*) | Strain: Swiss Webster (CFW) | Charles River | Strain Code: 024; RRID:IMSR_CRL:24 | |
| Antibody | Rabbit polyclonal anti-Aquaporin 1 | Santa Cruz Biotechnology | Cat# sc-20810; RRID:AB_2056824 | IF (1:250) |
| Antibody | Mouse monoclonal anti-Aquaporin 1 | Santa Cruz Biotechnology | Cat# sc-25287; RRID:AB_626694 | IF (1:250) |
| Antibody | Goat polyclonal anti-Aquaporin 2 | Santa Cruz Biotechnology | Cat# sc-9882; RRID:AB_2289903 | IF (1:250) |
| Antibody | Goat polyclonal anti-Aquaporin 3 | Santa Cruz Biotechnology | Cat# sc-9885; RRID:AB_2227514 | IF (1:250) |
| Antibody | Mouse monoclonal anti-Aquaporin 4 | Santa Cruz Biotechnology | Cat# sc-32739; RRID:AB_626695 | IF (1:250) |
| Antibody | Rabbit polyclonal anti-Aquaporin 5 | Santa Cruz Biotechnology | Cat# sc-28628; RRID:AB_2059871 | IF (1:250) |
| Antibody | Rat monoclonal anti-CD31 | Santa Cruz Biotechnology | Cat# sc-18916; RRID:AB_627028 | IF (1:250) |
| Antibody | Mouse monoclonal anti-CD164 | Santa Cruz Biotechnology | Cat# sc-271179; RRID:AB_10613973 | IF (1:250) |
| Antibody | Mouse monoclonal anti-Oligo1 | Santa Cruz Biotechnology | Cat# sc-166257; RRID:AB_2157524 | IF (1:250) |
| Antibody | Mouse monoclonal anti-Vimentin | Santa Cruz Biotechnology | Cat# sc-373717; RRID:AB_10917747 | IF (1:250) |
| Antibody | Rabbit polyclonal anti-GFAP | Abcam | Cat# ab7260; RRID:AB_10917747 | IF (1:500) |
| Antibody | Rabbit polyclonal anti-MRC1 | Abcam | Cat# ab64693; RRID:AB_1523910 | IF (1:500) |
| Antibody | Goat polyclonal anti-OMP | WAKO | Cat# 544–10001-WAKO; RRID:AB_664696 | IF (1:500) |
| Antibody | Mouse monoclonal anti-NeuN | Millipore | Cat# MAB377; RRID:AB_2298772 | IF (1:250) |
| Chemical compound, drug | DiI | Invitrogen | Cat# D282 | |
| Chemical compound, drug | Rhodamine Phalloidin | Invitrogen | Cat# R415 | (1:1000) |

*Continued on next page*

*Continued*

| Reagent type (species) or resource | Designation | Source or reference | Identifiers | Additional information |
|---|---|---|---|---|
| Chemical compound, drug | Albumin–fluorescein isothiocyanate conjugate (FITC-albumin) | Sigma | Cat# A9771 | |
| Chemical compound, drug | Formic Acid | Sigma | Cat# F0507 | |
| Chemical compound, drug | Fructose | Sigma | Cat# F0127 | |
| Chemical compound, drug | Kaolin | Sigma | Cat# K7375 | |
| Chemical compound, drug | α-thioglycerol | Sigma | Cat# M1753 | |
| Chemical compound, drug | Dextran, Fluorescein, 3000 MW, Anionic, Lysine Fixable (3kDa-FITCDextran) | ThermoFisher Scientific | Cat# D3306 | |
| Chemical compound, drug | Zinc Sulfate ($ZnSO_4$) | ThermoFisher Scientific | Cat# AC389802500 | |
| Chemical compound, drug | aCSF | Tocris Bioscience | Cat# 3525 | |
| Software, algorithm | Wheel Manager Software | MedAssociates | RRID:SCR_014296 | |

The protocols used in this study were approved by the Institutional Animal Care and Use Committee (IACUC) at the Pennsylvania State University. Mice were maintained on a 12 hr light-dark cycle with ad libitum access to food and water. We used both male and female Swiss Webster (Charles River, Wilmington, MA), C57BL/6J (Jackson Laboratory, Bar Harbor, ME), and LYVE1-GFP or LYVE1-tdtomato mice. We used a total of 294 mice [60–120 days/29.0 ± 5.8 g]. Partitioning was: Male/female (137/157), Swiss Webster (239), C57BL/6J (9) and LYVE1 mice (52). All anatomy studies were repeated in at least three mice. Mice used were ages 60–90 days (young adult). 'LYVE1-GFP' mice are the heterozygous offspring of $Lyve1^{EGFP-hCre+/+}$ (Jackson Laboratory - Stock No: 012601) and Ai6 (RCL-ZsGreen)$^{+/+}$ (Jackson Laboratory - Stock No: 007906) mice. 'LYVE1-tdtomato' mice are the heterozygous offspring of $Lyve1^{EGFP-hCre+/+}$ (Jackson Laboratory - Stock No: 012601) and Ai14 (Gt (ROSA)26Sor$^{tm14(CAG-tdTomato)Hze})^{+/+}$ (Jackson Laboratory - Stock No: 007914). This breeding scheme was used because the basal fluorescence of GFP in the $Lyve1^{EGFP-hCre}$ mice is low (*Pham et al., 2010*). No differences between the sexes were observed, so males and females were pooled. Randomization was used for all animal grouping.

## Histology

Mice were sacrificed via isoflurane overdose and perfused intracardially with heparinized-saline followed by 4% paraformaldehyde, unless otherwise noted. In some cases, the vasculature was labeled with perfusion of dye, FITC-albumin (MilliporeSigma, St. Louis, MO) or DiI (ThermoFisher Scientific, Waltham, MA) using the protocols described in (*Tsai et al., 2009*; *Ke et al., 2013*), respectively. The heads were fixed in 4% paraformaldehyde for 24 hr, then decalcified for 48 hr in formic acid (4%) solution, and saturated in 30% sucrose for sectioning. Tissue sections of 100 micrometers thick were sectioned on a freezing microtome. Sections were then either stained for immunofluorescence or thionin-nissl. For immunofluorescence, primary antibodies (and their respective dilutions) used on tissue sections were as follows: OMP (WAKO Chemicals U.S.A., Richmond, VA, 1:500), NeuN (MilliporeSigma, 1:250), Aquaporin-1 (Santa Cruz Biotechnology, Dallas, TX, 1:250), Aquaporin-2 (Santa Cruz, 1:250), Aquaporin-3 (Santa Cruz, 1:250), Aquaporin-4 (Santa Cruz, 1:250), Aquaporin-5 (Santa Cruz, 1:250), Oligo1 (Santa Cruz, 1:250), CD164 (Santa Cruz, 1:250), MRC1 (Abcam, Cambridge, United Kingdom, 1:500), CD31 (Santa Cruz, 1:250) and GFAP (Abcam, 1:500). Phalloidin conjugated to rhodamine was used at a 1:1000 dilution (Invitrogen). The following secondary antibodies were purchased from Abcam and used at a 1:500 working dilution: Goat Anti-Rabbit IgG H and L (Alexa

Fluor 488), Donkey Anti-Goat IgG H and L (Alexa Fluor 488), Goat Anti-Mouse IgG H and L (Alexa Fluor 488), Donkey Anti-Rabbit IgG H and L (Alexa Fluor 647), Donkey Anti-Goat IgG H and L (Alexa Fluor 647), and Goat Anti-Mouse IgG H and L (Alexa Fluor 647) preabsorbed. All stainings were done in 24 well plates, with one section per well. Sections were first blocked in either 4% goat or donkey serum for 1 hr at room temperature, then incubated in a 4% serum/primary antibody solution overnight at 4°C. Sections were then incubated in a 4% serum/secondary antibody solution for 1 hr at room temperature. Sections were mounted on silane-coated unifrost slides (Azer Scientific, Morgantown, PA), then coverslipped using fluoroshield mounting medium with DAPI (Abcam). Imaging was done on an Olympus Fluoview 1000 confocal, and images were processed in ImageJ (NIH). For thionin-nissl staining, sections were first mounted on silane-coated unifrost slides and allowed to dry overnight in a 37°C incubator. Slides were then immersed in 95% EtOH, 70% EtOH, 50% EtOH, distilled water, thionin-nissl staining solution, 50%, 70% EtOH, 95% EtOH, 100% EtOH, and xylene and coverslipped using Cytoseal (Thermofisher Scientific).

## Cisterna magna (CM) injections

The animal was anesthetized with isoflurane (5% induction, 2% maintenance) in oxygen. The dorsal surface of the head was elevated 8 mm above the back of the animal and the head was tilted downward at an angle of 10 degrees from horizontal. The base of the skull was exposed by an incision, and a fine glass-pipette (~20 micrometers in diameter for dye injections) or a 26G needle (for kaolin injections) was inserted into the cisterna magna using a stereotaxic instrument (Stoelting, Wood Dale, IL). Either 2.0 µL of 2% Evans blue in aCSF (Tocris Bioscience, Bristol, United Kingdom) or 5.0 µL of 10% solution of 3kDa-FITCDextran in aCSF was infused at a constant rate of 0.2 µL/min using a syringe pump (Harvard Apparatus, Holliston, MA), an infusion rate that does not raise ICP in mice (*Yang et al., 2013*). After the injection, the animal was sacrificed via isoflurane overdose and perfused for histology ten minutes after the end of the infusion of dye. For kaolin injections, twenty microliters of 250 mg/mL kaolin (MilliporeSigma) in aCSF (Tocris Bioscience) was infused at a constant rate of 5.0 µL/min using a syringe pump (Harvard Apparatus). After the injection, the incision was sutured and the animal was monitored for 2 days until ICP measurements were made.

## Olfactory bulb injections

The animal was anesthetized with isoflurane in oxygen (5% induction, 2% maintenance). The dorsal surface of the skull was exposed by an incision, and a fine glass-pipette was inserted into the right olfactory bulb using a stereotaxic instrument (Stoelting, Wood Dale, IL). One microliter of 5% solution of 3kDa-FITCDextran in aCSF was infused at a constant rate of 0.1 µL/min using a syringe pump (Harvard Apparatus, Holliston, MA), an infusion rate that does not raise ICP in mice (*Yang et al., 2013*). After the injection, the animal was sacrificed via isoflurane overdose and perfused for histology 10 min after the end of the infusion of dye.

## SeeDB clearing of the spinal column

A modified version of the SeeDB protocol (*Ke et al., 2013*; *Hartmann et al., 2015*) was used to clear the spinal column. The spinal columns were decalcified in formic acid (4%) solution for two days. Next, the spinal columns were saturated in a fructose and α-thioglycerol gradient: 20%–8 hr, 40%–12 hr, 60%–24 hr, 80%–24 hr, 100%–48 hr, and SeeDB solution – 72 hr. Once cleared, the dorsal side of the spinal columns were imaged under a light microscope.

## CT imaging of calcified tissue

Swiss Webster skulls were imaged with a GE v|tome|x L 300 high-resolution nano/microCT scanner and C57BL/6J skulls were imaged on an OMNI-X HD600 industrial microCT scanner (Varian Medical Systems, Palo Alto, CA). Scan settings were 100kV, 0.1mA (GE) or 160kV, 0.25 mA. Image stacks were processed and analyzed in Avizo 8. For each skull, a threshold that segmented the calcified and non-calcified tissue was chosen such that the zygotic maxillary suture, which is externally visible by eye, was clearly defined in the segmented image. To validate that the CT images accurately captured the fine details of the CP, in a subset of skulls the cranial vault was removed using a Dremel tool and the CP photographed on a Zeiss StereoDiscovery V8.

## Intranasal administration of substances

After the mouse had been rendered unconscious by a brief exposure to isoflurane, 20 μL of $ZnSO_4$ (10% in sterile $H_2O$) or vehicle control (sterile $H_2O$) was administered to the left nare with a blunt 32G needle. The animal was then inverted to allow for excess fluid to exit the nasal cavity. The animals were monitored and weighed daily after treatment. Approximately 4% of animals treated with $ZnSO_4$ died within 24 hr of administration.

## Quantification of movement of Evans blue (EB) and olfactory bulb size

Images were obtained of the turbinates and olfactory bulb post-mortem in midline sagittally-sectioned $ZnSO_4$- and vehicle-treated mice using a Zeiss StereoDiscovery V8 with a 2000R scientific CCD camera (QImaging, Surrey, BC Canada) utilizing μManager software (*Edelstein et al., 2014*). We used the red channel, in which EB dye appears as a decrease in intensity, for all analyses. To determine analogous sections between vehicle control and $ZnSO_4$-treated animals, the anatomy of the hippocampus was used since olfactory bulb degeneration is observed after $ZnSO_4$ treatment (*Burd, 1993*). To measure the dye movement into the nasal epithelium, a rectangular ROI (5 mm rostral-caudal and 3 mm dorsal-ventral in extent) was placed with the caudal-dorsal corner located at the inflection point of the dorsal surface of the olfactory bulb. The center of mass, the location of maximal dye concentration (peak) along the rostral caudal axis, and the maximal dye value were quantified and plotted using Matlab. Using ImageJ, olfactory bulb ROIs were manually drawn around the outer edge of the olfactory bulb to quantify olfactory bulb size. The concentration of albumin in CSF is approximately 300 mg/L, (at 70 kDa/mole, this works out to $\sim 4*10^{-6}$ moles/liter) and there are 35 microliters CSF in the mouse (implying $\sim 1.4*10^{-10}$ moles of albumin total). There are 12–14 Evans blue (Evans blue: 1 mg*0.02/ (960g/mole)=$2*10^{-8}$ moles Evans blue) binding sites per albumin, so $\sim 10\%$ of the Evans blue molecules injected will be bound to CSF albumin (*Freedman and Johnson, 1969*; *Ganrot and Laurell, 1974*).

For quantifying dye movement along the spinal column, spinal columns were separated from the skull between the C2 and C3 vertebrae, cleared in SeeDB reagents, and photos of the dorsal side of the spinal columns were taken using a Zeiss StereoDiscovery V8 with a 2000R scientific CCD camera (QImaging) utilizing μManager software (*Edelstein et al., 2014*) (*Figure 9D* top). Using ImageJ, the image was split into red, green, and blue channels, and only the red channel was used for analysis (*Figure 9D* bottom). To compensate for the small bends present in the fixed spinal columns, for each spinal column, a line (yellow) was manually drawn along the midline, from the anterior to posterior end (*Figure 9D* bottom). The dye intensity was measured along this line as a function of distance traveled down the spinal column (*Figure 9F*).

## RT-PCR

RNA was extracted from tissue using the Trizol and chloroform method (described in *Simms et al., 1993*), and converted to cDNA (Thermo Scientific Verso cDNA synthesis Kit). Primers designed in Primer blast for *Klotho,* and *AQP-1*, validated on the kidney, were used to determine mRNA expression levels. Primer sequences: *AQP1*, 5'CATGGTGGAAGCCAGTTCCTT (FOR) and 5'CAGAC-CAGGGTGTGTAGTGG (REV); *Klotho*, 5'GTGGCCGAGAGAGTTTTGGA (FOR) and 5'GGGGTC TCACCTTTCAGAGC (REV). Primers for *NF-KB* and *RelA* were previously validated (*Yamamoto et al., 2009*). *NF-KB1*: 5'GAAATTCCTGATCCAGACAAAAAC (FOR) and 5' A TCACTTCAATGGCCTCTGTGTAG (REV). *RelA:* 5'CTTCCTCAGCCATGGTACCTCT (FOR) and 5'CAAGTCTTCATCAGCATCAAACTG (REV). Mouse *GAPDH* was used as a house-keeping gene and loading control: 5'AGGCCGGTGCTGAGTATGTC (FOR) and 5' TGCCTGCTTCACCACCTTCT (REV) (ShineGene Molecular Biotech, Inc).

## Intracranial pressure (ICP) measurements

All surgical procedures were under isoflurane anesthesia (5% for induction and 2% for maintenance). A titanium head-bar was attached to the skull with cyanoacrylate glue and dental cement (*Drew et al., 2010*; *Shih et al., 2012*) and the skull was covered with a thin layer of cyanoacrylate glue. After 2 days of recovery, the animal was habituated to head fixation on a spherical treadmill for 1 day (for three 30 min sessions). On the day of the ICP experiment (1 day after the habituation), the mouse was anesthetized with isoflurane and a small craniotomy (~1 mm diameter)

was made in the somatosensory cortex. A pressure-measuring catheter (SPR-1000, Millar, Houston, TX) was inserted into the cortex ($-1.0$ mm dorsal, $-1$ mm ventral from bregma), and a tight seal was made using Qwik-sil (World Precision Instruments, Sarasota, FL). This surgical procedure took approximately 10 min. The animal was allowed to wake from anesthesia and to freely locomote on the spherical treadmill (*Huo et al., 2014*; *Huo et al., 2015*) for 2 hr, during which both ICP and locomotion were recorded simultaneously at 1 kHz (NI USB-6003). To minimize any residual effect of anesthesia on ICP (*Gao and Drew, 2016*), we only analyzed data collected more than 1 hr after the cessation of anesthesia for resting ICP data. The first 2 min after surgery and cessation of isoflurane, when the animal was stationary, were used for the 'rest after isoflurane' analysis.

## Statistical analysis

Unless noted, all statistical analysis was performed using Matlab (R2015b, MathWorks, Natick, MA), with the Matlab function used (i.e. 'ttest2') is listed. All summary data were reported as the mean ± standard deviation (SD). Normality (Anderson-Darling test, adtest) and of the samples were tested before statistical testing. When the data conform to the normality assumption, difference between treatments were compared using unpaired t-test (ttest2). If the condition of normality was not met, parametric tests (ttest2) were substituted with a non-parametric method (Mann-Whitney U test, ranksum). For comparing the effects of treatment and duration on CSF drainage, two-way ANOVA was used. For the wheel activity data, the K-S test was used. For the ICP data, a mixed model ANOVA test was used. A result was considered significant if $p \leq 0.05$ after correcting for multiple comparisons using the Bonferonni correction.

# Acknowledgements

This work was supported by a Scholar Award from the McKnight Endowment Fund for Neuroscience, National Institutes of Health Grant R01NS078168, P01HD078233, and NSF grant CBET1705854 to PJD, and a Huck Graduate Enrichment Grant and F31NS105461 from the NIH to JNN. We thank B Strowbridge for suggesting $ZnSO_4$, A Shih and D Hartman for assistance with SeeDB, J Richtsmeier and members of the Drew lab for comments on the manuscript. All confocal imaging was done using the Penn State Microscopy and Cytometry Facility - University Park, PA.

# Additional information

## Funding

| Funder | Grant reference number | Author |
| --- | --- | --- |
| National Institutes of Health | F31NS105461 | Jordan N Norwood |
| Pennsylvania State University | Huck Graduate Enrichment Grant | Jordan N Norwood |
| National Science Foundation | CBET1705854 | Patrick J Drew |
| McKnight Endowment Fund for Neuroscience | | Patrick J Drew |
| National Institutes of Health | R01NS078168 | Patrick J Drew |
| National Institutes of Health | P01HD078233 | Patrick J Drew |

The funders had no role in study design, data collection and interpretation, or the decision to submit the work for publication.

## Author contributions

Jordan N Norwood, Conceptualization, Formal analysis, Funding acquisition, Investigation, Visualization, Methodology, Writing—original draft, Writing—review and editing; Qingguang Zhang, Formal analysis, Investigation, Visualization, Methodology; David Card, Formal analysis, Investigation, Methodology; Amanda Craine, Timothy M Ryan, Investigation, Methodology; Patrick J Drew, Conceptualization, Software, Formal analysis, Supervision, Funding acquisition, Writing—original draft, Project administration, Writing—review and editing

## Author ORCIDs
Jordan N Norwood  http://orcid.org/0000-0001-8093-5938
Qingguang Zhang  http://orcid.org/0000-0003-4500-813X
Patrick J Drew  https://orcid.org/0000-0002-7483-7378

## Ethics
Animal experimentation: The protocols used in this study were approved by the Institutional Animal Care and Use Committee (IACUC) at the Pennsylvania State University

## Decision letter and Author response
Decision letter https://doi.org/10.7554/eLife.44278.024
Author response https://doi.org/10.7554/eLife.44278.025

## Additional files

### Supplementary files
• Transparent reporting form
DOI: https://doi.org/10.7554/eLife.44278.022

### Data availability
All raw data is plotted in the figures. ICP data and code (Figure 10) is included in a zip file. Code for the analysis of actograms is available at https://github.com/DrewLab/MedAssociates_WheelActivity (copy archived at https://github.com/elifesciences-publications/MedAssociates_WheelActivity).

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
