## [Decision Letter]

Thank you for submitting your article "Anatomical basis and physiological role of cerebrospinal fluid transport through the murine cribriform plate" for consideration by *eLife*. Your article has been reviewed by Ronald Calabrese as the Senior Editor, a Reviewing Editor, and three reviewers. The following individuals involved in review of your submission have agreed to reveal their identity: Roxana Carare (Reviewer #1); Robert G Thorne (Reviewer #2); Jeff Illif (Reviewer #3).

The reviewers have discussed the reviews with one another and the Reviewing Editor has drafted this decision to help you prepare a revised submission.

Summary:

This is a nicely executed study with some very interesting and novel findings related to olfactory nerve – cribriform plate (CP) – olfactory mucosa relationships. Overall this study provides important anatomical insight into an understudied region of the skull base which has recently re-emerged interest-wise into the neuro-immunology field owing to the recent detailed characterization of meningeal lymphatic vessels by the Alitalo and Kipnis groups. Arguably the strongest portion of the work relates to the anatomical descriptions of the nerves/vessels/lymphatics/water channels in this key interface region, previously not well described in the rodent. Somewhat weaker are the author's ZnSO_4_ nerve ablation experiments because of the difficulty / challenge in interpreting the resulting findings. Still there is a lot of new compelling data here and a novel way of looking at the CSF system.

Essential revisions:

The reviewers were in agreement that the portion of the manuscript dealing with CSF secretion needs to be extensively rewritten. It should be possible for the authors to address the comments that all the reviewers have made while retaining much of the content, not only the anatomical description portion. Specifically, for the functional studies, the authors must account much better for other possibilities and not be so definitive in their conclusions. Particularly the conclusion that lesioning the olfactory path in some way results in a feed-back regulation of CSF secretion. The experiment performed simply isn't solid enough to definitively answer that question. The other functional question – the contribution that drainage down this pathway makes to overall drainage along the cervical lymphatics helps to place the study within the wider experimental context of the field (both historical and current elements), but this experiment is not totally definitive either and must be discussed with care.

Reviewer #1:

This is an elegant study encompassing novel methodology and demonstrating firstly the exact anatomical features of the cribriform plates and their penetrating structures. Secondly, the authors demonstrate that ablating the olfactory sensory neurons results in the decreased production of CSF with ICP remaining unchanged. My main question is whether the authors could be more specific about the nature of the vessels that express LYVE1- are they arteries or veins and is LYVE1 present also on capillaries.

Reviewer #2:

This is a nicely executed study with some very interesting and novel findings related to olfactory nerve – cribriform plate (CP) – olfactory mucosa relationships. Arguably the strongest portion of the work relates to the anatomical descriptions of the nerves/vessels/lymphatics/water channels in this key interface region, previously not well described in the rodent. Somewhat weaker are the author's ZnSO_4_ nerve ablation experiments because of the obvious difficulty / challenge in interpreting the resulting findings. The authors would be wise to revisit some of their conclusions, softening the description and better allowing for alternatives (e.g. statements such as 'one possibility may be that…' and 'although further work is needed to definitively conclude X, we think it likely that…' would greatly help their manuscript stand the test of time and not be seen by some specialists in the field as over-reaching in places). Nevertheless, the results are important. So long as some of the interpretations acquire a bit more nuance and care in the strength of conclusions derived from them, the experiments described in this manuscript represent a significant advance for the rapidly growing research area concerned with lymphatic drainage from the brain.

1) The authors would do well to comment upon and relate their findings to Buck and Axel's 2004 Nobel Prize winning studies (superfamily of GPCR odorant receptors) and subsequent related work better identifying zones and projection patterns from the olfactory epithelium to glomeruli in the bulb. For example, do the authors believe that these projections are organized in any special way between the major and smaller foramina that they describe in the manuscript? What do the authors think are the primary differences between the major foramina and smaller, minor foramina that are so consistently represented in the two strains of mice they examined? This is an excellent topic for speculation that the authors somewhat avoid in favor of less wise discussion later in the manuscript over their ablation experiments (see below). However, the possible significance of the anatomical relationships they describe more fully for the first time in the mouse would certainly seem to merit a bit more discussion.

2) Subsection “Blood vessels traverse the cribriform plate”. The authors state in the results that 'blood vessels traverse the CP in the mouse, much like the ethmoidal arteries observed in humans (refs), linking the vascular territories of the nasal epithelia and brain.' This statement would seem to imply that the ethmoidal artery – CP relationship has not been described in mice / rodents but there are publications that previously discuss this (e.g., Lochhead and Thorne, 2012). Of some interest and not commented upon is venous drainage. Did the authors identify veins traversing the CP? Is there any link between draining veins of the olfactory mucosa and the olfactory sinus via the CP? The authors should at minimum comment on this, citing what is known from the literature, so that their manuscript provides more complete coverage of the subject area. It is not beyond the scope of what they have studied and described in the manuscript to more fully on these important points. The vascular relationships between the nasal mucosae and brain are quite important clinically; better elaboration of these relationships in the rodent will aid in understanding the relevance (and possible limitations) of rodent models of disease.

3) Subsection “Aquaporins at the olfactory nerve-bulb interface”. AQP1 is described as lining the 'foramina of the CP along the crista galli.' Several questions arise: did they mean to imply that it is only the smaller foramina and not the major foramina that had this AQP1 expression? Looking at Shields et al., 2010 that identified olfactory ensheathing cells as expressing AQP1, it is not clear if the authors agree that their findings are consistent with that reference or that their AQP1 expression is rather localized to the olfactory nerve fibroblasts that surround axon bundles (and would naturally be in a location lining the outside of foramina; it has been well described that ensheathing cells wrap individual axon bundles and would be more uniformly represented in nerve bundle cross sections, rather than at the peripheral edges; see Kumar et al., 2018 Figure 5E and associated references). The authors should further comment on this. Most importantly, Weller et al., have described in a recent publication (2018) the expression of AQP1 on 'dural border cells' of the meninges with some discussion about the CP area (e.g., in Figure 1D and F of the 2018 paper). The authors of the current manuscript describe AQP1 as limited to the CP area they describe and the choroid plexus, but this description lacks the real complexity of the fuller picture described in the Weller et al. reference and some of Mollgard's prior related work. The authors should consult this literature and more fully elaboration their AQP discussion.

4) The authors argue in their ablation experiment description (subsection “Ablation of OSNs decreases CSF outflow”) that 'because the space between the OSN axons provides a conduit for outflow of CSF and ISF, removing these axons should decrease outflow of CSF.' However, the opposite might be argued as well. If axons are removed from bundles surrounded by olfactory ensheathing cells, then the space may be filled in, so to speak, by fluid secretion, thus creating effectively larger volume paths for fluid outflow. Perhaps this might be an alternative explanation for the observation that EB outflow to the deep cervical lymph nodes was unchanged in ZnSO_4_-treated animals and for the lack of ICP increase they observed? The authors would do well to flesh out their interpretation in this section of the manuscript far better, allowing for other possibilities. It would seem to be easily argued that the authors results do NOT definitively 'show that in response to the disruption of the nasal outflow pathway… CSF production was decreased' (subsection “OSN ablation does not affect ICP”), as the authors assert in the closing sentence, simply because CSF production was not directly measured. Inference is quite different from proof! The authors would do well to recognize the countless number of times in science where results looked straightforward in allowing a certain inference at a given time, only to be made to look naïve with subsequent future studies showing a more complex situation. Accordingly, I strongly advise that the statements and related discussion by the authors regarding interpretation of the ZnSO_4_ ablation experiments really need to better reflect this, more fully allowing for other interpretations and possibilities, not least that the physiology may be much more complicated than their description allows for (see comments in opening summary). Such a revision will not require a major change or rewrite, only the addition of a few more carefully and thoughtfully placed qualifiers and perhaps some further thought about alternatives. The Discussion section accomplishes this balancing act better than the Results section, as currently written. The Results section would be greatly improved by making this strongly advised change.

Reviewer #3:

The study by Norwood and colleagues uses several imaging approaches to define the anatomical landscape of the cribriform plate and nasal turbinates; then using fluorescent tracers and chemical ablation of the olfactory nerves, they demonstrate that this ablation appears to reduce CSF efflux across the cribriform plate, reportedly associated with changes in CSF secretion. Overall this study provides important anatomical insight into an understudied region of the skull base which has recently re-emerged interest-wise into the neuro-immunology field owing to the recent detailed characterization of meningeal lymphatic vessels by the Alitalo and Kipnis groups. While the anatomical descriptions of these barriers appears to be conducted well and with care, and are themselves a useful addition to the field, the functional elements of the study that seek to address what this efflux pathway is doing, and what the consequences of its lesioning are, are generally underdeveloped and should be strengthened to improve the value of the study to the field. Most critical items follow:

Effect of nerve lesion on CSF secretion and reabsorption. By plotting CSF tracer distribution along the neuroaxis, the authors note that distribution shifts away from the anterior portion with olfactory nerve ligation, although distribution down the spinal cord is also reduced. From this, a reduction in CSF secretin is inferred, supported by unchanged ICP measurements in awake-behaving animals.

- A reduced CSF secretion would be very interesting. CSF secretion should be evaluated directly using a gold-standard ventriculo-cisternal tracer perfusion method.

- ICP monitoring was conducting in waking, moving animals, while tracer studies were conducted in anesthetized animals. Either ICP monitoring should be done under the same physiological state as the tracer studies, or vise-versa.

- The authors note no change in the lymphatic vasculature at the cribriform plate with Zn ablation. But that tracer movement across is reduced. Is overall CSF drainage down the cervical lymphatic trunks altered? If so, then this would support their contention that altering cribriform plate efflux alters CSF dynamics, while no change in CSF tracer efflux along the lymphatic trunks would argue that compensatory efflux along another route is occurring.

- Lastly – does the nerve lesion result in any neuroinflammation – either glial changes with the parenchymal compartment, or immune cell changes in the meningeal or CSF compartments? Such changes might be predicted to alter CSF tracer dynamics and distribution and should be ruled in or out.

Tracer uptake into lymphatic vessels. The presence of lymphatic vessels at the cribriform plate is described, and the results of the present study clearly are relevant to how meningeal lymphatic vessels throughout the cranium, including in the skull base and in association with dural sinuses function – what their relative contribution to CSF solute efflux are, and how they are impacted by the intervention to lesion olfactory nerves. Some questions remain on this that are important for placing the significance of this efflux pathway relative to these others.

- Do CSF tracers concentrate in the cribriform plate lymphatic vessels? Or is there tracer simply permeating this structure, which also has lymphatic vessels within it? Do the authors note uptake of tracer into dural sinus-associated lymphatic vessels (again – concentration within them, not simply labeling of meninges).

- Does uptake into non-cribriform lymphatic vessels change with nerve lesion? This would be very interesting question to answer regardless of the outcome and would be related to overall lymphatic efflux issue noted above.

---

## [Author Response]

Summary:This is a nicely executed study with some very interesting and novel findings related to olfactory nerve – cribriform plate (CP) – olfactory mucosa relationships. Overall this study provides important anatomical insight into an understudied region of the skull base which has recently re-emerged interest-wise into the neuro-immunology field owing to the recent detailed characterization of meningeal lymphatic vessels by the Alitalo and Kipnis groups. Arguably the strongest portion of the work relates to the anatomical descriptions of the nerves/vessels/lymphatics/water channels in this key interface region, previously not well described in the rodent. Somewhat weaker are the author's ZnSO_4_ nerve ablation experiments because of the difficulty / challenge in interpreting the resulting findings. Still there is a lot of new compelling data here and a novel way of looking at the CSF system.Essential revisions:The reviewers were in agreement that the portion of the manuscript dealing with CSF secretion needs to be extensively rewritten. It should be possible for the authors to address the comments that all the reviewers have made while retaining much of the content, not only the anatomical description portion. Specifically, for the functional studies, the authors must account much better for other possibilities and not be so definitive in their conclusions. Particularly the conclusion that lesioning the olfactory path in some way results in a feed-back regulation of CSF secretion. The experiment performed simply isn't solid enough to definitively answer that question. The other functional question – the contribution that drainage down this pathway makes to overall drainage along the cervical lymphatics helps to place the study within the wider experimental context of the field (both historical and current elements), but this experiment is not totally definitive either and must be discussed with care.

In response to the reviewer’s comments, we removed the text claiming that ZnSO_4_ treatment causes a decrease in CSF production. We have added text to the Discussion section discussing other possible explanations, and a statement that our experiments cannot distinguish between the possible origins of lack of an ICP increase. The Discussion section now reads:

“Interestingly, instead of a rise in ICP as would be expected due to a blockage of CSF drainage into the nasal cavity, we observed no change in ICP. This could not be explained by an increase in drainage of CSF down the spinal cord pathway, as we observed a similar reduction in CSF flow down this pathway following ZnSO_4_ treatment. We can think of two non-mutually exclusive possibilities that could explain the lack of ICP rise when an outflow pathway is blocked. The first is that other drainage pathways become engaged and/or reduce their outflow resistance to compensate for the blockage of the cribriform plate outflow. In addition to providing enough drainage to prevent ICP rise, the CSF flow dynamics would be altered by these outflow resistance changes such that the flow down the spinal cord is decreased. One possible location for outflow increase this could by the lymphatics at the skull base (Antila et al., 2017) or along the other nerve bundles that exit the skull (Bradbury and Cole, 1980; Kida, Pantazis and Weller, 1993; Kida et al., 1994). A re-routing of CSF flow in a more rostral direction would account for the decreased flow down the spinal cord. Some outflow pathways have thresholds for opening (Welch and Friedman 1960), such as arachnoid villi, and these might become engaged following ZnSO_4_ treatment. While arachnoid villi have been described in primates and other species, there are no definitive anatomical studies in the mouse. The second possibility could be a decrease in the production of CSF. CSF production is not a passive process, but rather one under active neural control (Lindvall, Edvinsson and Owman, 1978; Edvinsson and Lindvall, 1978; Tuor et al., 1990). As ICP needs to be maintained within a healthy range, there are likely homeostatic processes that sense ICP, and cause compensatory increases or decreases in CSF production, similar to other mechanosensory processes in other organs of the body (Umans and Liberles, 2018; Sarah-Jane et al., 2018). A potential mechanism mediating this feedback is mechanically-sensitive ASIC3 channels, which are expressed in the ventricles (Jalalvand et al., 2018), and are sensitive to changes in pressure that are physiologically relevant to ICP regulation. Pathological elevations of ICP due to trauma produce a rise in systemic blood pressure and a decrease in heart rate, known as the Cushing response (Hoff and Reis, 1970; Doba and Reis, 1972), showing the brain has a way of sensing ICP (Hoff and Reis, 1970), and pressure-sensitive brain regions mediating these responses have been identified (Doba and Reis, 1972; Hoff and Reis, 1970). A similar response may be occurring in the choroid plexus to compensate for the decreased CSF drainage we observe after ZnSO_4_ treatment, thus, avoiding the rise in ICP that is normally associated with a decrease in CSF drainage (Lindvall, Edvinsson and Owman, 1978; Edvinsson and Lindvall, 1978). Neither of these two possibilities excludes the other, and from the experiments performed here, we are unable to differentiate between these two possibilities.”

Reviewer #1:This is an elegant study encompassing novel methodology and demonstrating firstly the exact anatomical features of the cribriform plates and their penetrating structures. Secondly, the authors demonstrate that ablating the olfactory sensory neurons results in the decreased production of CSF with ICP remaining unchanged. My main question is whether the authors could be more specific about the nature of the vessels that express LYVE1- are they arteries or veins and is LYVE1 present also on capillaries.

To answer this question, we stained brain sections from LYVE1-Ai6 mice with rhodamine-conjugated phalloidin. We observed LYVE1 expression along both arteries and capillaries in the mouse cortex. Results of these experiments were added to Figure 3—figure supplement 2 and the subsection “Lymphatic vessels traverse the cribriform plate”.

Reviewer #2:This is a nicely executed study with some very interesting and novel findings related to olfactory nerve – cribriform plate (CP) – olfactory mucosa relationships. Arguably the strongest portion of the work relates to the anatomical descriptions of the nerves/vessels/lymphatics/water channels in this key interface region, previously not well described in the rodent. Somewhat weaker are the author's ZnSO_4_ nerve ablation experiments because of the obvious difficulty / challenge in interpreting the resulting findings. The authors would be wise to revisit some of their conclusions, softening the description and better allowing for alternatives (e.g. statements such as 'one possibility may be that…' and 'although further work is needed to definitively conclude X, we think it likely that…' would greatly help their manuscript stand the test of time and not be seen by some specialists in the field as over-reaching in places). Nevertheless, the results are important. So long as some of the interpretations acquire a bit more nuance and care in the strength of conclusions derived from them, the experiments described in this manuscript represent a significant advance for the rapidly growing research area concerned with lymphatic drainage from the brain.1) The authors would do well to comment upon and relate their findings to Buck and Axel's 2004 Nobel Prize winning studies (superfamily of GPCR odorant receptors) and subsequent related work better identifying zones and projection patterns from the olfactory epithelium to glomeruli in the bulb. For example, do the authors believe that these projections are organized in any special way between the major and smaller foramina that they describe in the manuscript? What do the authors think are the primary differences between the major foramina and smaller, minor foramina that are so consistently represented in the two strains of mice they examined? This is an excellent topic for speculation that the authors somewhat avoid in favor of less wise discussion later in the manuscript over their ablation experiments (see below). However, the possible significance of the anatomical relationships they describe more fully for the first time in the mouse would certainly seem to merit a bit more discussion.

The question regarding the anatomical relationships and patterns of the projections through the major and minor foramina could be addressed by using markers (either genetic or immunohistochemically) for specific olfactory receptor proteins and histologically determining if the axons of a specific receptor type exhibited a stereotyped pattern or preference for foramina. We did not check for any such relationships or patterns, but have added the following to the Discussion section:

“We observed a highly stereotyped pattern in the positioning of the major foramina across mice. It is tempting to speculate that the OSN axons are predisposed to go through either the dorsal or ventral foramina depending on their receptor group and target glomeruli (Buck and Axel, 1991; Feinstein and Mombaerts, 2004; Rodriguez-Gil et al., 2010).”

2) Subsection “Blood vessels traverse the cribriform plate”. The authors state in the results that 'blood vessels traverse the CP in the mouse, much like the ethmoidal arteries observed in humans (refs), linking the vascular territories of the nasal epithelia and brain.' This statement would seem to imply that the ethmoidal artery – CP relationship has not been described in mice / rodents but there are publications that previously discuss this (e.g., Lochhead and Thorne, 2012).

We thank the reviewer for bringing this literature to our attention. The relationship of the vascular territories of the nasal epithelia and brain were already very well documented in (Lochhead and Thorne, 2012a). This comparison was used to emphasize the similarities of the anatomy of this area between humans and mice and the usefulness and applicability of using the mouse as a model for this system. We have added the Lochhead and Thorne reference and other references, so the section now reads:

“These results are consistent with previous work showing blood vessels traverse the CP in the rodent (Lochhead and Thorne, 2012b), much like the ethmoidal arteries and veins observed in humans (Knudsen, Andersen and, Krag 1989; Yang et al., 2009; Souza et al., 2009; Tsutsumi, Ono and Yasumoto, 2016; Tsutsumi et al., 2019; Lochhead and Thorne, 2012b), linking the vascular territories of the nasal epithelia and brain.”

Of some interest and not commented upon is venous drainage. Did the authors identify veins traversing the CP? Is there any link between draining veins of the olfactory mucosa and the olfactory sinus via the CP? The authors should at minimum comment on this, citing what is known from the literature, so that their manuscript provides more complete coverage of the subject area. It is not beyond the scope of what they have studied and described in the manuscript to more fully on these important points. The vascular relationships between the nasal mucosae and brain are quite important clinically; better elaboration of these relationships in the rodent will aid in understanding the relevance (and possible limitations) of rodent models of disease.

Whether the vessels traversing the cribriform plate were arteries or veins cannot be definitively concluded from our experiments. The vessels were identified by FITC-albumin or DiI cardiovascular fills, neither of which provides any contrast between arteries and veins. The section on the cribriform plate traversing vessels now reads:

“We first looked at whether blood vessels traversed the CP by either filling the vessel lumen with FITC-albumin (Tsai et al., 2009), or labeling the endothelial cells by perfusing the vasculature with the fluorescent lipophilic dye DiI (Li et al., 2008). Note that we cannot distinguish between arteries or veins using these methods.”

3) Subsection “Aquaporins at the olfactory nerve-bulb interface”. AQP1 is described as lining the 'foramina of the CP along the crista galli.' Several questions arise: did they mean to imply that it is only the smaller foramina and not the major foramina that had this AQP1 expression?

We did not observe any AQP1 expression lining the major foramina of the cribriform plate, and have clarified the relevant section to read:

“We observed high levels of AQP1 expression on the olfactory nerve layer, down to the glomeruli, and in the lamina propria of the neuroepithelium at both the medial (Figure 2F-G) and dorsal (Figure 2H-K) junctions of the olfactory bulb and nerve, but not lining the major foramina.”

Looking at Shields et al., 2010 that identified olfactory ensheathing cells as expressing AQP1, it is not clear if the authors agree that their findings are consistent with that reference or that their AQP1 expression is rather localized to the olfactory nerve fibroblasts that surround axon bundles (and would naturally be in a location lining the outside of foramina; it has been well described that ensheathing cells wrap individual axon bundles and would be more uniformly represented in nerve bundle cross sections, rather than at the peripheral edges; see Kumar et al., 2018 Figure 5E and associated references). The authors should further comment on this.

The localization of expression and pattern of staining of AQP1 along the olfactory bulb observed would support previous findings that olfactory ensheathing cells along the periphery of the olfactory bulb express AQP1 (Shields et al., 2010). Expression of AQP1 in fibroblasts was checked by staining with Vimentin (expressed by fibroblasts) in the foramina of the cribriform plate but co-localization of AQP1 and vimentin expression was not observed. Based on these results, we believe the cells expressing AQP1 that surround the axon bundles might be olfactory ensheathing cells (OECs). We have added figures showing AQP1 and Vimentin antibody staining in the cribriform plate to Figure 2—figure supplement 1J-K.

Text below was added to the Results section:

“We observed high levels of AQP1 expression on the olfactory nerve layer down to the glomeruli, as observed previously (Shields et al., 2010), and in the lamina propria of the neuroepithelium at both the medial (Figure 2F-G) and dorsal (Figure 2H-K) junctions of the olfactory bulb and nerve. AQP1 was also observed lining the foramina of the CP along the crista galli (Figure 2L, N-O) but did not co-label with Vimentin-positive fibroblasts (Figure 2—figure supplement 1J-K). Based on previous reports (Au, Treloar and Greer, 2002; Kumar et al., 2018) and our own results, we hypothesize that these cells might be olfactory ensheathing cells (OECs).”

Most importantly, Weller et al., have described in a recent publication (2018) the expression of AQP1 on 'dural border cells' of the meninges with some discussion about the CP area (e.g., in Figure 1D and f of the 2018 paper). The authors of the current manuscript describe AQP1 as limited to the CP area they describe and the choroid plexus, but this description lacks the real complexity of the fuller picture described in the Weller et al. reference and some of Mollgard's prior related work. The authors should consult this literature and more fully elaboration their AQP discussion.

Reviewer #2 brings up a good point about AQP1 expression in development that we overlooked. Embryonically, AQP1 expression begins around E13 (in the rat) and its pattern and sequence of development is maintained throughout many mammals, including humans (Johansson et al., 2005). Interestingly, AQP1 expression was also observed in dural border cells of the embryonic rat dura (Weller et al., 2018). Text below was added to the Results section:

“We first looked at the expression of aquaporin-1 (AQP1), as it has been shown that expression in the CNS begins early (Johansson et al., 2005; Weller et al., 2018), and is required in adulthood to maintain CSF production by the choroid plexus (Oshio et al., 2005). Interestingly, previous work has also shown that AQP1 is expressed along the periphery of the olfactory bulb in neonatal mice (Shields et al., 2010), and there is a high level of expression of AQP1, 3, and 5, within the nasal cavity (Ablimit et al., 2006).”

4) The authors argue in their ablation experiment description (subsection “Ablation of OSNs decreases CSF outflow”) that 'because the space between the OSN axons provides a conduit for outflow of CSF and ISF, removing these axons should decrease outflow of CSF.' However, the opposite might be argued as well. If axons are removed from bundles surrounded by olfactory ensheathing cells, then the space may be filled in, so to speak, by fluid secretion, thus creating effectively larger volume paths for fluid outflow. Perhaps this might be an alternative explanation for the observation that EB outflow to the deep cervical lymph nodes was unchanged in ZnSO_4_-treated animals and for the lack of ICP increase they observed?

There are two possible outcomes from our ablation experiment: either ablation of the OSNs creates a hole, or the removal of the OSNs leads to sealing of the opening. Since we did not observe flow into the nasal cavity, there is unlikely to be a hole. Additionally, larger volume paths were not observed, since there were large decreases in EB and fluorescein in the nasal cavity after ZnSO_4_ treatment (Figure 7 and Figure 7—figure supplement 1). An alternative explanation to the unchanged EB outflow to the deep cervical lymph nodes is that the EB dye may be entering other outflow pathways at the skull base, such as the optic nerves or lymphatics in the spinal column (see response to “Essential revisions”).

The authors would do well to flesh out their interpretation in this section of the manuscript far better, allowing for other possibilities. It would seem to be easily argued that the authors results do NOT definitively 'show that in response to the disruption of the nasal outflow pathway… CSF production was decreased' (subsection “OSN ablation does not affect ICP”), as the authors assert in the closing sentence, simply because CSF production was not directly measured. Inference is quite different from proof!

The reviewer brings up a very good point and we have addressed this above.

The authors would do well to recognize the countless number of times in science where results looked straightforward in allowing a certain inference at a given time, only to be made to look naïve with subsequent future studies showing a more complex situation. Accordingly, I strongly advise that the statements and related discussion by the authors regarding interpretation of the ZnSO_4_ ablation experiments really need to better reflect this, more fully allowing for other interpretations and possibilities, not least that the physiology may be much more complicated than their description allows for (see comments in opening summary). Such a revision will not require a major change or rewrite, only the addition of a few more carefully and thoughtfully placed qualifiers and perhaps some further thought about alternatives. The Discussion section accomplishes this balancing act better than the Results section, as currently written. The Results section would be greatly improved by making this strongly advised change.

We have rewritten the Discussion section to cover alternate possibilities (see response to “Essential revisions” above).

Reviewer #3:The study by Norwood and colleagues uses several imaging approaches to define the anatomical landscape of the cribriform plate and nasal turbinates; then using fluorescent tracers and chemical ablation of the olfactory nerves, they demonstrate that this ablation appears to reduce CSF efflux across the cribriform plate, reportedly associated with changes in CSF secretion. Overall this study provides important anatomical insight into an understudied region of the skull base which has recently re-emerged interest-wise into the neuro-immunology field owing to the recent detailed characterization of meningeal lymphatic vessels by the Alitalo and Kipnis groups. While the anatomical descriptions of these barriers appears to be conducted well and with care, and are themselves a useful addition to the field, the functional elements of the study that seek to address what this efflux pathway is doing, and what the consequences of its lesioning are, are generally underdeveloped and should be strengthened to improve the value of the study to the field. Most critical items follow:Effect of nerve lesion on CSF secretion and reabsorption. By plotting CSF tracer distribution along the neuroaxis, the authors note that distribution shifts away from the anterior portion with olfactory nerve ligation, although distribution down the spinal cord is also reduced. From this, a reduction in CSF secretin is inferred, supported by unchanged ICP measurements in awake-behaving animals.- A reduced CSF secretion would be very interesting. CSF secretion should be evaluated directly using a gold-standard ventriculo-cisternal tracer perfusion method.

While this has been used as a gold-standard for measuring CSF secretion, the results are often times variable, even in much larger animals than the mouse. Based on this variability and the challenging nature of this method in the mouse, we did not perform these experiments. We hope that future technical innovations make these experiments more feasible in a mouse model.

- ICP monitoring was conducting in waking, moving animals, while tracer studies were conducted in anesthetized animals. Either ICP monitoring should be done under the same physiological state as the tracer studies, or vise-versa.

The reviewer brings up a good point. In response to the reviewer’s point, we have added ICP measurements made at the end of the surgery in the two minutes after the cessation of isoflurane but before the animal regained consciousness (Figure 10D, Supplementary file 2). The concentration of isoflurane in the mouse during this period will still be very high. In general, the ICP was higher during this period, consistent with previous results that show isoflurane increases resting ICP (Gao and Drew, 2016). We find that there is no difference between the ICP of the different groups immediately after isoflurane. While this is not the optimal experiment, it is the best we can do without repeating this series of experiments in anesthetized animals. The text below was added to the Results section:

“No statistically significant difference was observed in ZnSO_4_ treated mice at rest or during locomotion as compared to the vehicle controls (Figure 10B-C) measured 10, 30, and 60 days after treatment (Supplementary file 2). Additionally, there was no difference between the groups’ ICP immediately after isoflurane cessation and before the animal regained consciousness (Figure 10D, Supplementary file 2).”

- The authors note no change in the lymphatic vasculature at the cribriform plate with Zn ablation. But that tracer movement across is reduced. Is overall CSF drainage down the cervical lymphatic trunks altered? If so, then this would support their contention that altering cribriform plate efflux alters CSF dynamics, while no change in CSF tracer efflux along the lymphatic trunks would argue that compensatory efflux along another route is occurring.

We were unable to quantitatively measure the Evans blue signal in the lymph nodes in the neck optically. Future experiments using radioactive tracers and collection of lymphatic fluid flow should be better positioned to quantify any changes.

- Lastly – does the nerve lesion result in any neuroinflammation – either glial changes with the parenchymal compartment, or immune cell changes in the meningeal or CSF compartments? Such changes might be predicted to alter CSF tracer dynamics and distribution and should be ruled in or out.

To address the reviewer’s question, we assayed glial changes in the cortex and inflammation marker mRNA expression in the choroid plexus both 2 and 10 days after ZnSO_4_ treatment. However, no changes in any of the inflammation markers was observed. This has been added to Figure 6—figure supplement 2, and the following was added to the Results section:

“We then assayed mRNA levels in the choroid plexus and GFAP expression in the cortex to look for immune responses to ZnSO_4_ treatment. AQP1, expressed by epithelial cells in the choroid plexus, is involved in CSF production (Oshio et al., 2005). Klotho is a protein involved in suppressing aging and is significantly elevated in the CSF of Alzheimer’s disease patients (Kuro-o et al., 1997; Semba et al., 2014). The NF-κβ pathway is involved in the choroid plexus acute-phase response to peripheral inflammation (Marques et al., 2009), and RelA is required for NF- κβ activation (Chen and Greene 2004). GFAP is a hallmark marker for astrocytes and is upregulated as part of an inflammatory response (Sofroniew, 2009). However, we found no appreciable changes in mRNA levels or GFAP expression (Figure 6 —figure supplement 2) 2 or 10 days after ZnSO_4_ treatment indicating there is no inflammatory response in the cranial cavity following treatment.”

Tracer uptake into lymphatic vessels. The presence of lymphatic vessels at the cribriform plate is described, and the results of the present study clearly are relevant to how meningeal lymphatic vessels throughout the cranium, including in the skull base and in association with dural sinuses function – what their relative contribution to CSF solute efflux are, and how they are impacted by the intervention to lesion olfactory nerves. Some questions remain on this that are important for placing the significance of this efflux pathway relative to these others.- Do CSF tracers concentrate in the cribriform plate lymphatic vessels? Or is there tracer simply permeating this structure, which also has lymphatic vessels within it?

We observe dye throughout the tissue in the major foramina, which has LYVE1^+^-vessels in it. Though the dye co-localizes within these vessels, it does not concentrate in it. We have clarified this in the text:

“3kDa-FITCDextran was observed draining into the nasal cavity along the olfactory nerve (Figure 5D-F), and co-localizing with LYVE1+ vessels along the olfactory nerve (Figure 5G-I), though the dye did not appear to be concentrated in the LYVE1+ vessels.”

Do the authors note uptake of tracer into dural sinus-associated lymphatic vessels (again – concentration within them, not simply labeling of meninges).- Does uptake into non-cribriform lymphatic vessels change with nerve lesion? This would be very interesting question to answer regardless of the outcome and would be related to overall lymphatic efflux issue noted above.

In preliminary experiments in untreated mice, we did not see cisterna magna-injected dye in the dural sinus-associated lymphatic vessels or other locations in the dura using our dye-infusion protocol. We did not check the dural sinus-associated lymphatic vessels in ZnSO_4_-treated mice. The presence of the dye in the dural likely depends on the specifics of the injection and perfusion protocol (timing, etc.), so we are hesitant to generalize our results.